# Sensory-memory interactions via modular structure explain errors in visual working memory

**Jun Yang[1†, ‡], Hanqi Zhang[2,3,4], Sukbin Lim[2,3,4]\***

[1]Weiyang College, Tsinghua University, Beijing, China; [2]Shanghai Frontiers Science Center of Artificial Intelligence and Deep Learning, Shanghai, China; [3]Neural Science, Shanghai, China; [4]NYU-ECNU Institute of Brain and Cognitive Science, Shanghai, China

**\*For correspondence:**
sukbin.lim@nyu.edu

**Present address:**
[†]Interdisciplinary Graduate Program in Quantitative Biosciences, Georgia Institute of Technology, Atlanta, United States; [‡]School of Mathematics, Georgia Institute of Technology, Atlanta, United States

**Competing interest:** The authors declare that no competing interests exist.

**Abstract** Errors in stimulus estimation reveal how stimulus representation changes during cognitive processes. Repulsive bias and minimum variance observed near cardinal axes are well-known error patterns typically associated with visual orientation perception. Recent experiments suggest that these errors continuously evolve during working memory, posing a challenge that neither static sensory models nor traditional memory models can address. Here, we demonstrate that these evolving errors, maintaining characteristic shapes, require network interaction between two distinct modules. Each module fulfills efficient sensory encoding and memory maintenance, which cannot be achieved simultaneously in a single-module network. The sensory module exhibits heterogeneous tuning with strong inhibitory modulation reflecting natural orientation statistics. While the memory module, operating alone, supports homogeneous representation via continuous attractor dynamics, the fully connected network forms discrete attractors with moderate drift speed and nonuniform diffusion processes. Together, our work underscores the significance of sensory-memory interaction in continuously shaping stimulus representation during working memory.

## eLife assessment

This **important** computational study provides new insights into how neural dynamics may lead to time-evolving behavioral errors as observed in certain working-memory tasks. By combining ideas from efficient coding and attractor neural networks, the authors construct a two-module network model to capture the sensory-memory interactions and the distributed nature of working memory representations. They provide **convincing** evidence supporting that their two-module network, although none of the alternative circuit structures they considered can account for error patterns reported in orientation-estimation tasks with delays.

## Introduction

The brain does not faithfully represent external stimuli. Even for low-level features like orientation, spatial frequency, or color of visual stimuli, their internal representations are thought to be modified by a range of cognitive processes, including perception, memory, and decision (*Geisler, 2008*; *Webster, 2015*; *Bays et al., 2022*). Experimental studies quantified such modification by analyzing behavior data or decoding neural activities. For instance, biases of errors, the systematic deviation from the original stimuli, observed in estimation tasks have been used as indirect evidence to infer changes in the internal representations of stimuli (*Wei and Stocker, 2017*).

One important source of biases is adaptation to environmental statistics, such as the nonuniform stimulus distribution found in nature or the limited range in specific settings. Cardinal repulsion, which refers to the systematic shift away from the horizontal and vertical orientations observed in many perceptual tasks, is one of the examples (*de Gardelle et al., 2010*). Theoretical works suggest that such a bias pattern reflects the prevalence of the cardinal orientations in natural scenes (*Girshick et al., 2011*). Similarly, the variance of errors for orientation stimuli was found to be inversely proportional to the stimulus statistics, minimum at cardinal and maximum at oblique orientations (*van Bergen et al., 2015*). It was postulated that the dependence of biases and variance of errors on natural statistics results from sensory encoding optimized to enhance precision around the most common stimuli (*Ganguli and Simoncelli, 2014*; *Wei and Stocker, 2015*; *Wei and Stocker, 2017*).

On the other hand, there is a growing body of evidence indicating that error patterns are not solely influenced by sensory encoding but are also shaped by memory processes. In delayed estimation tasks, where participants are presented with stimuli followed by a delay period during which they rely on their working memory for estimation, it has been observed that representations of orientation or color stimuli undergo gradual and continuous modifications throughout the delay period (*Panichello et al., 2019*; *Bae, 2021*; *Gu et al., 2023*). Such dynamic error patterns are inconsistent with sensory encoding models, most of which only establish a static relationship between stimuli and internal representations.

Traditional working memory models are not suitable either. Most of them are constructed to faithfully maintain information about stimuli during the delay period, and thus, the memory representation has a similar geometry as that of the stimuli (*Wang, 2001*; *Khona and Fiete, 2022*). For continuous stimuli such as orientation, location, direction, or color, all stimuli are equally maintained in ring-like memory activities, predicting no biases (*Zhang, 1996*; *Compte et al., 2000*; *Burak and Fiete, 2009*).

How can we explain error patterns in working memory tasks that are similar to those observed in perception tasks? Here, we claim that not a single-module but a two-module network with recursive interaction is required. Each module has a distinct role – sensory encoding and memory maintenance. To illustrate this, we use orientation stimuli and examine how their representations change during the delayed estimation tasks. We employ two approaches to find solutions for generating correct error patterns. The first extends previously suggested sensory encoding models, while the second modifies low-dimensional memory models based on attractor dynamics. These approaches are integrated into the network models, which link network connectivity to neuronal tuning properties and behavioral error patterns and reveal the attractor dynamics through low-dimensional projection. Our results show that the sensory-memory interacting networks outperform single-module networks with better control over the shapes and evolution of dynamic error patterns. Furthermore, our network models emphasize the importance of inhibitory tuning in sensory circuits for generating correct error patterns under typical associative learning of natural statistics. Finally, we provide testable predictions regarding the effect of perturbations in sensory-memory interactions on error patterns in delayed estimation tasks.

## Results

### Low-dimensional attractor models

In natural images, cardinal orientations are the most prevalent (*Figure 1A*). Error patterns in estimation tasks show dependence on such natural statistics, such as biases away from cardinal orientations where the variance of errors is nonetheless minimal (*Figure 1B and C*). In delayed estimation tasks, such a bias pattern is consolidated in time (*Figure 1B*). Also, experimental data suggested that estimation errors increase with a longer delay (*Wimmer et al., 2014*; *Schneegans and Bays, 2018*), while the precision is still highest at cardinal orientations (*van den Berg et al., 2012*; *Bays, 2014*; *van Bergen et al., 2015*). Thus, we assumed that the variance of errors increases as keeping its characteristic shape (*Figure 1C*). To explain these errors across orientations and over time, we first explored the underlying working memory mechanism. We considered low-dimensional attractor models with input noise that describe the drift and diffusion of the memory states. Here, we show that two prominent classes of previously suggested models are inconsistent with experimental observations and examine what modification to the models is required.

The most widely accepted model for working memory of orientation stimuli has continuous attractor dynamics, which assumes that all orientations are equally encoded and maintained (*Figure 1D–F*).

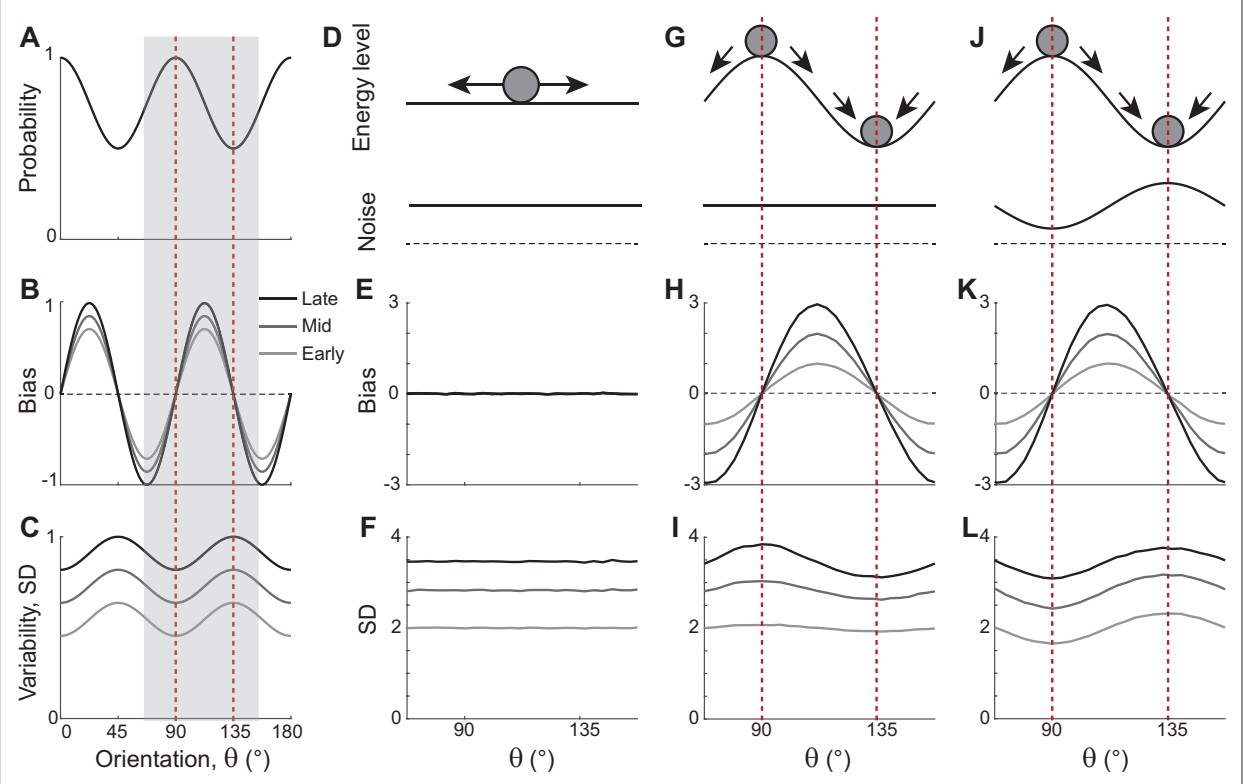

**Figure 1.** Error patterns of orientation stimuli in delayed estimation tasks and low-dimensional attractor models. (**A–C**) Characteristic patterns of natural statistics of orientation stimuli $\theta$ (**A**), bias (**B**), and standard deviation (SD; **C**) during the delay period observed experimentally. Cardinal orientations are predominant in natural images (**A**). Bias and SD increase during the delay period, keeping patterns of repulsive bias (**B**) and minimum variance (**C**) around cardinal orientations. These characteristic patterns are visualized using trigonometric functions, and the range is normalized by their maximum values. Red vertical lines correspond to representative cardinal and oblique orientations, and with a periodicity of the error patterns, we only show the gray-shaded range in the remaining panels. (**D–L**) Comparison of different attractor models. (**D–F**) Continuous attractors with constant noise. Energy potential is flat (**D**), resulting in no bias (**E**) and uniform SD with uniform noise (**F**). (**G–L**) Discrete attractors with constant (**G–I**) and nonuniform noise (**J–L**). The discrete attractor models have potential hills and wells at cardinal and oblique orientations, respectively (**G, J**). While the bias patterns depend only on the energy landscape (**H, K**), SD representing variability also depends on noise (**I, L**). For the correct SD pattern (**L**), uneven noise with its maxima at the obliques (**J**) is required. Bias and SD patterns in the attractor models were obtained by running one-dimensional drift-diffusion models (see Methods).

Each attractor corresponds to the memory state for different stimuli and forms a continuous ring following the geometry of orientation stimuli. The dynamics along continuous attractors are conceptually represented as movement along a flat energy landscape (*Figure 1D*). Without external input, there is no systematic shift of mean activity, i.e., no drift during the delay period (*Figure 1E*). Also, under the assumption of equal influence of noise for all orientations, the variance of errors is spatially flat with constant diffusion along the ring, while the overall magnitude increases over time due to the accumulation of noise (*Figure 1F*).

While such continuous attractor models have been considered suitable for memory storage of continuous stimuli, they cannot capture drift dynamics observed during the delay period. Instead, discrete attractor models with uneven energy landscapes have been suggested with the energy wells corresponding to discrete attractors (*Figure 1G–I*). As evolution toward a few discrete attractors creates drift dynamics, the bias increases during the delay (*Figure 1H*). Also, discrete attractor models naturally produce nonuniform variance patterns. Even with constant noise along the ring, variance becomes minimum/maximum at the attractors/repellers due to the drift dynamics (*Figure 1I*). However, discrete attractor models with constant noise yield inconsistent results when inferring the locus of attractors from the bias and variance patterns observed in the data. Cardinal orientations should be the repeller to account for cardinal repulsion. In contrast, the minimum variance observed at the cardinal orientations suggests they should be the attractors.

How can such inconsistency be resolved? One possible solution is discrete attractor models with nonuniform noise amplitude (*Figure 1J*). Let's consider that attractors are formed at oblique orientations to generate correct bias patterns (*Figure 1K*). Additionally, we assumed that noise has the highest amplitude at the obliques. When the difference in the noise amplitude is large enough to overcome the attraction toward the obliques, the models can produce correct variance patterns, maximum at the obliques and minimum at cardinal orientations (*Figure 1L*). In sum, unlike two prominent memory models, continuous attractors or discrete attractors with constant noise, discrete attractors with maximum noise at the obliques could reproduce experimentally observed error patterns of orientation stimuli. Note that these attractor models often simplify the full network dynamics. Namely, the drift and diffusion terms are derived by projecting network dynamics onto low-dimensional memory states (*Burak and Fiete, 2012*; *Darshan and Rivkind, 2022*). Thus, it is still in question whether there exist memory networks that can implement attractor dynamics with correct drift and diffusion terms.

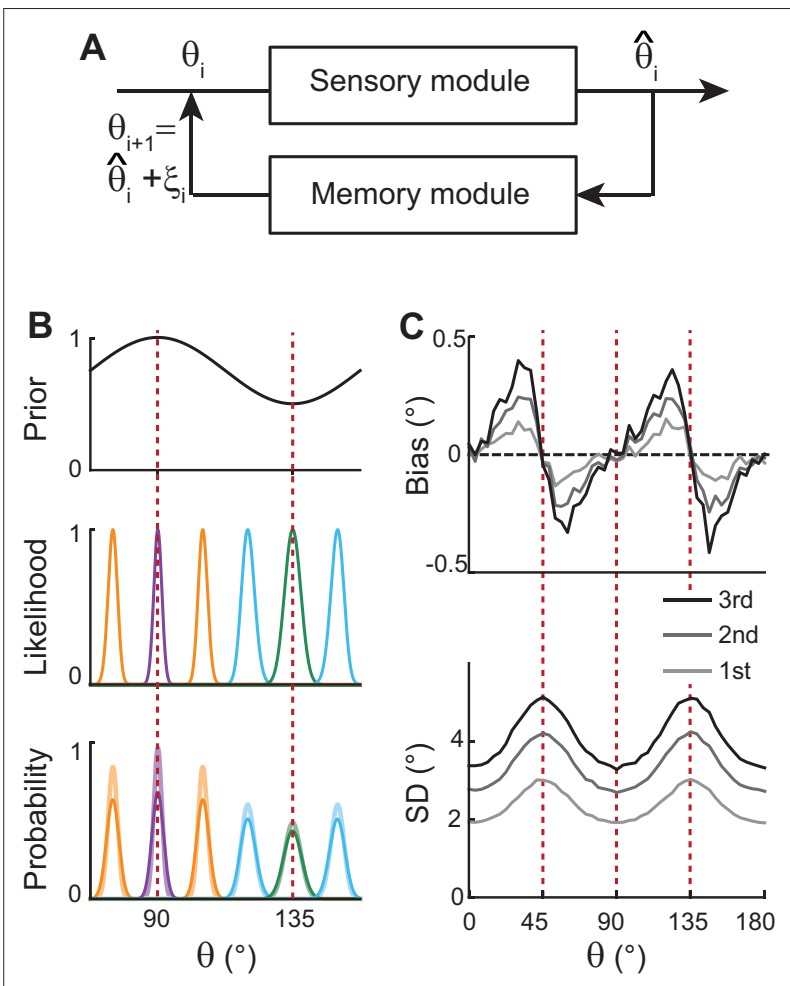

**Figure 2.** Extension of Bayesian sensory models. (**A**) Schematics of extension to memory processing. We adapted the previous Bayesian models (*Wei and Stocker, 2015*) for sensory encoding where $\theta$ and $\hat{\theta}$ are the input and output of sensory modules, respectively. We added a memory module where it maintains $\hat{\theta}$ with the addition of memory noise $\xi$. The output of the memory module, $\hat{\theta} + \xi$, is fed back to the sensory module as the input for the next iteration. (**B**) Illustration of the first iteration of sensory-memory interaction. Prior distribution follows the natural statistics (top), resulting in a sharper likelihood function near cardinal orientations (middle). Combining prior and likelihood functions leads to the posterior distribution of decoded $\hat{\theta}$ (light colors at the bottom), which is broadened with the addition of memory noise (dark colors at the bottom). Different curves correspond to different initial $\theta$. (**C**) Bias (top) and SD (bottom) patterns obtained from decoded $\hat{\theta}$ for the first, second, and third iterations.

## Bayesian sensory model and extension

Before exploring full memory network models, we note that previous theoretical works for sensory processing suggested that Bayesian inference with efficient coding could generate the repulsive bias and the lowest variance at cardinal orientations (*Wei and Stocker, 2015*; *Wei and Stocker, 2017*). Efficient coding theory suggests the sensory system should enhance the sensitivity around more common stimuli. For orientation stimuli, precision should be highest around cardinal directions, which could be achieved by sharpening the likelihood functions. Equipped with Bayesian optimal readout, such a sensory system could reproduce correct error patterns observed in perceptual tasks for various visual stimuli, including orientations (*Figure 2*).

However, such models only account for the relationship between external and perceived stimuli during sensory processing, resulting in static error patterns. Here, we extended the framework so that the system can maintain information about the stimulus after its offset while bias and variance of errors grow in time (*Figure 2A*). We added a memory stage to Bayesian sensory models such that the memory stage receives the output of the sensory stage and returns it as the input after the maintenance. For instance, let's denote the external orientation stimulus given during the stimulus period as $\theta_1$. The sensory stage receives $\theta_1$ as input and generates the perceived orientation, $\hat{\theta}_1$, which varies from trial to trial with sensory noise (*Figure 2B*). Through the memory stage, $\hat{\theta}_1$ is returned as the input to the sensory stage for the next iteration with the addition of memory noise $\xi_1$.

Such a recursive process mimics interactions between sensory and memory systems where the sensory system implements efficient coding and Bayesian inference, and the memory system faithfully maintains information. As the recursive process iterates, the distribution of the internal representation of orientation broadens due to the accumulation of noise from the sensory and memory systems. This leads to an increase in bias and variance at each step while keeping their characteristic shapes (*Figure 2C*). Thus, recurrent interaction between sensory and memory systems during the delay period, each of which meets different demands, successfully reproduces correct error patterns observed in memory tasks.

## Network models with sensory and memory modules

Next, we construct network models capturing the sensory-memory interactions formalized under the Bayesian framework. We consider two-module networks where each module corresponds to the sensory and memory systems. To generate orientation selectivity, both modules have a columnar architecture where neurons in each column have a similar preference for orientation (*Figure 3A*). However, their connectivity structures are different (*Figure 3B*). The memory module in isolation resembles the traditional ring attractor network with a strong and homogeneous recurrent connection. This enables the memory module in isolation to maintain information about all orientations equally during the delay period (*Figure 3B–F*, right). Conversely, the recurrent connectivity strengths in the sensory module are relatively weak, such that without connection to the memory module, the activities during the delay period decay back to the baseline levels (*Figure 3B*, left). Furthermore, the connectivity strengths across columns are heterogeneous, particularly stronger at the obliques. As a result, the tuning curves near cardinal orientations can be sharper and denser, consistent with experimental observations showing a larger number of cardinally tuned neurons (*Li et al., 2003*; *Shen et al., 2014*) and their narrower tuning (*Li et al., 2003*; *Kreile et al., 2011*; *Figure 3C–F*, left). Different response activities of the two modules in isolation are demonstrated in their response manifolds as more dispersed representations around cardinal orientations in the sensory module, compared to the ring-like geometry of the memory module (*Figure 3F*).

For sensory-memory interacting networks, we connected the two modules with intermodule connections set to be stronger between neurons with similar orientation selectivity (*Figure 4A*). Activity profiles in both modules follow that of the sensory module – heterogeneous with narrower and denser tuning curves around cardinal orientations, leading to higher sensitivity (*Figure 4B*). Such activity pattern is maintained even during the delay period when recurrent connections in the memory module support activities of both sensory and memory modules (*Figure 4B*, right). Note that while sensory activities convey stimulus information during the delay period, their overall firing rates are much lower than those during the stimulus period with weak interconnection strengths. Such low firing rates may lead to both positive and negative evidence of sustained activity in early sensory areas (*Leavitt et al., 2017*).

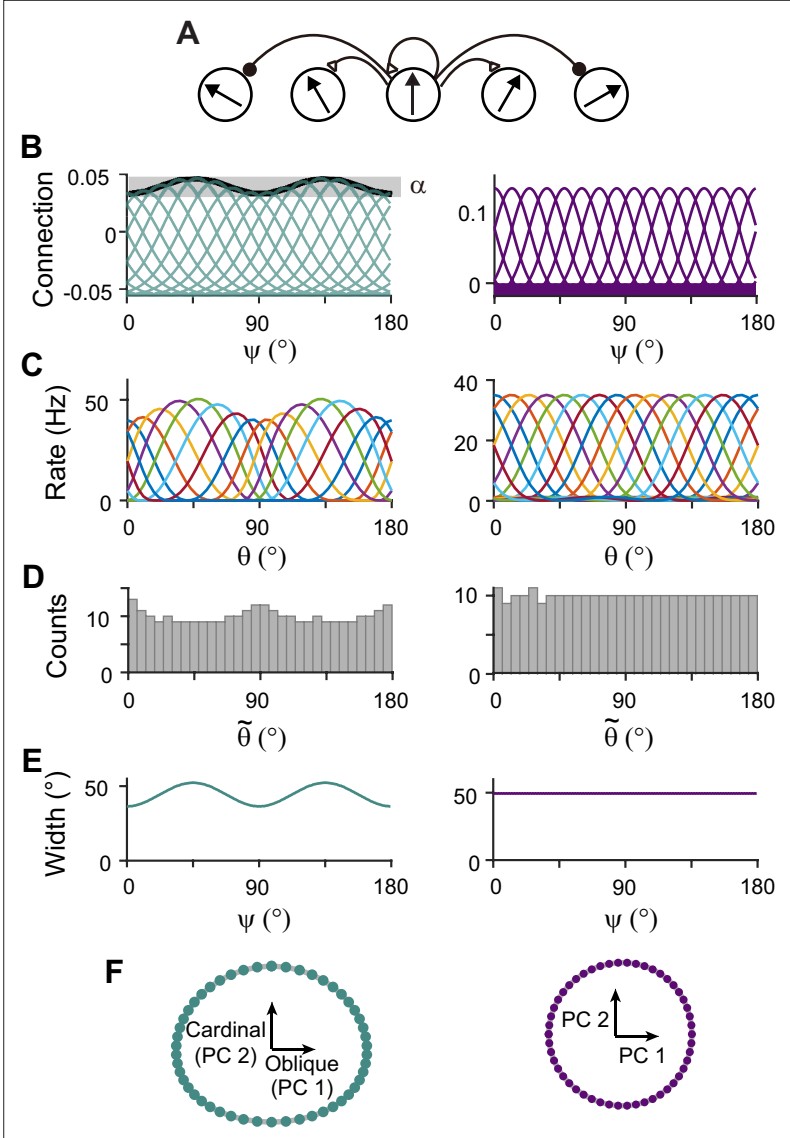

**Figure 3.** Network models of sensory and memory circuits in isolation, implementing efficient coding and ring attractor dynamics, respectively. (**A**) Schematics of columnar architecture for orientation selectivity. Neurons in the same column have similar preferred orientations, and recurrent connections are a combination of local excitation and global inhibition, represented as triangles and circles, respectively. (**B–F**) Connectivity and tuning properties of the sensory network (left column) and memory network (right column). (**B**) Example connectivity strengths. We indexed neurons by $\psi$ ranging uniformly from 0° to 180°. The connectivity strengths depend only on $\psi$'s of the presynaptic and postsynaptic neurons. Each curve shows the connectivity strengths from presynaptic neuron $\psi$ to an example postsynaptic neuron. Unlike the homogeneous connectivity in the memory network (right), the sensory connectivity is heterogeneous, and its degree is denoted by $\alpha$. (**C**) Heterogeneous tuning curves for different stimulus $\theta$ in the sensory network in the stimulus period (left) and homogeneous ones in the memory network in the delay period (right). The memory network can sustain persistent activity in isolation, while the sensory network cannot. (**D**) Histograms of the preferred orientations. We measured the maximum of the tuning curve of each neuron, denoted as $\tilde{\theta}$ (Methods). The heterogeneous sensory network has more cardinally tuned neurons. (**E**) Widths of tuning curves measured at the half maximum of the tuning curves (Methods). The sensory tuning curves sharpen around cardinal orientations. Each neuron is labeled with its index $\psi$ as in (**B**). (**F**) Neural manifolds projected onto the first two principal components of activities during the stimulus period (left) and during the delay period (right). The neural manifold of the sensory network resembles a curved ellipsoid, while the manifold corresponding to the homogeneous memory network is a perfect ring.

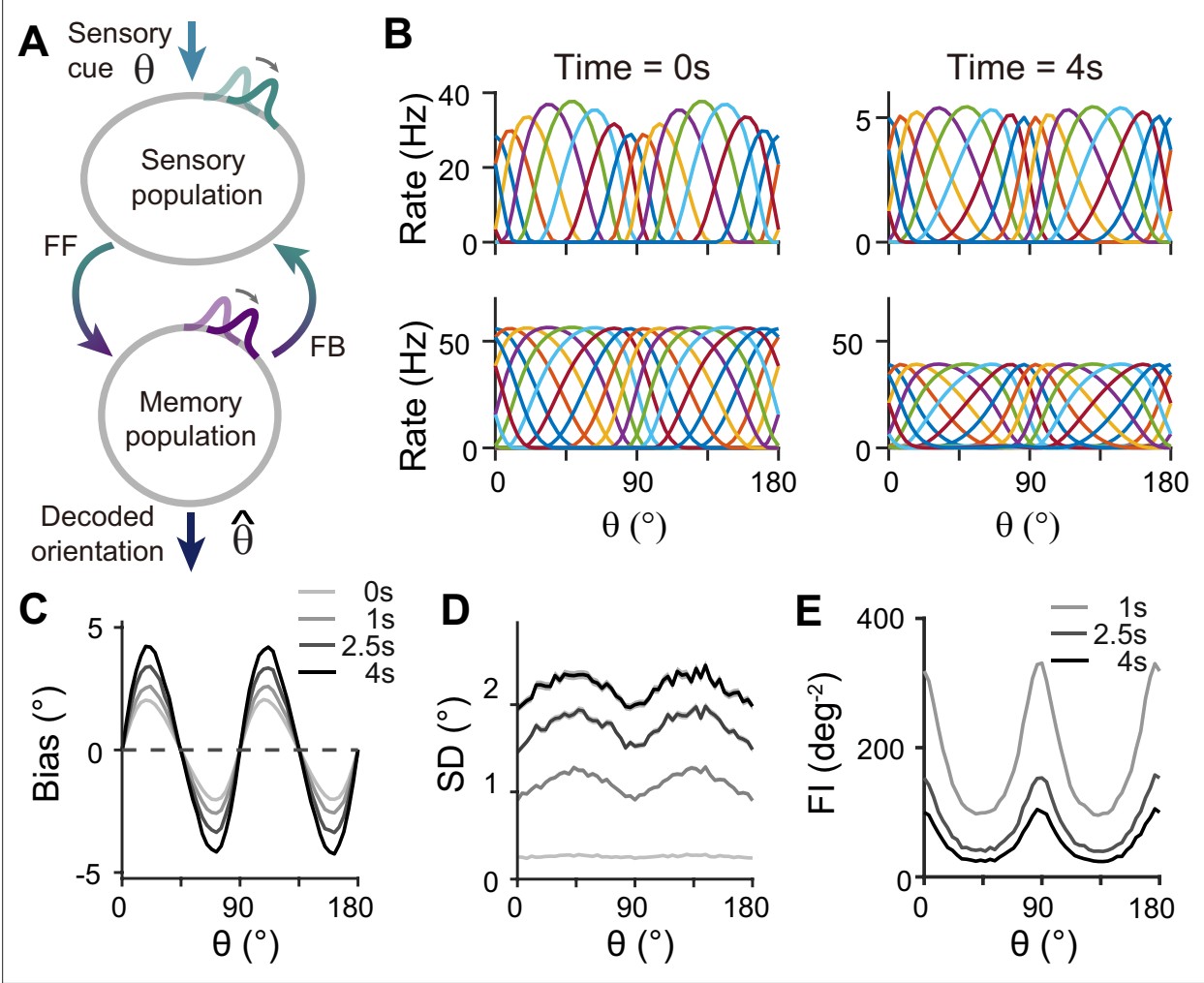

**Figure 4.** Network model with interacting sensory and memory modules generates correct error patterns in delayed estimation tasks. (**A**) Schematic of two-module architecture. The sensory and memory modules are connected via feedforward and feedback connectivity to form a closed loop. The sensory module receives external input with orientation $\theta$ while internal representation is decoded from the memory module, denoted as $\hat{\theta}$. (**B**) Tuning curves of sensory (upper panels) and memory (lower panels) modules at the end of the stimulus epoch (i.e. the beginning of the delay epoch; left panels) and during the delay period (right panels). Note that while both modules can sustain persistent activity in the delay period, the firing rates of the sensory module are significantly lower than those in the stimulus period (upper right). (**C–E**) Bias (**C**), standard deviation (SD; **D**), and Fisher information (FI; **E**) patterns. Error patterns evaluated at 1, 2.5, and 4 s into the delay are consistent with the characteristic patterns observed experimentally in delayed estimation tasks (*Figure 1A–C*). However, the low SD right after the stimulus offset in (**D**) deviates from error patterns seen in perception tasks (see Discussion). While FI decays due to noise accumulation, it is largest around cardinal orientations, corresponding to a smaller discrimination threshold (**E**). In (**C**) and (**D**), shaded areas mark the ±s.e.m. of 1000 realizations.

The online version of this article includes the following figure supplement(s) for figure 4:

**Figure supplement 1.** Bias (**A**) and SD (**B**) patterns decoded from activities of sensory module.

**Figure supplement 2.** Dynamics of bias and tuning properties of sensory-memory interacting network models.

When the internal representation of the orientation stimulus is read from the memory module using a population vector decoder mimicking Bayesian optimal readout (*Fischer, 2010*), the sensory-memory interacting network exhibits repulsive bias and minimum variance at cardinal orientations, inheriting from efficient sensory coding (*Figure 4C and D*). Similar error patterns were observed when decoded from activities of the sensory module (*Figure 4—figure supplement 1*). Such bias increases during the delay period with increasing asymmetry of tuning widths despite lower firing rates than the stimulus period (*Figure 4—figure supplement 2*). At the same time, errors gradually increase due to noise accumulation in time, as in typical memory networks (*Compte et al., 2000*; *Burak and Fiete, 2012*). Note that the variance of errors is negligible during stimulus presentation when the

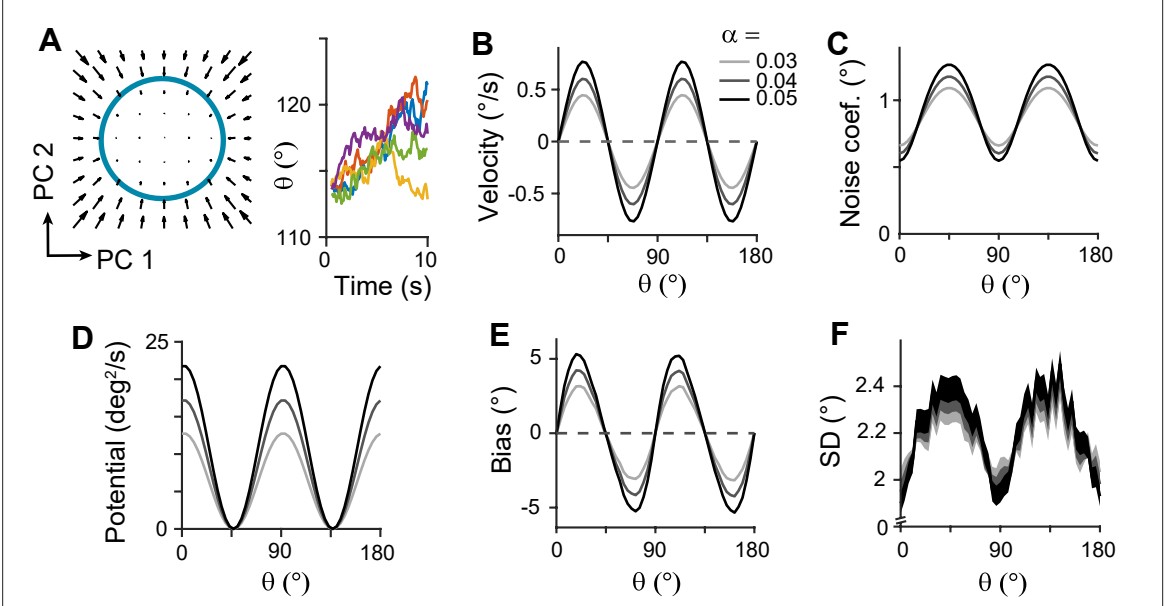

**Figure 5.** Low-dimensional dynamics along memory manifold and their dependence on heterogeneity degrees in the sensory module. (**A**) Low-dimensional projection along the memory states. Left panel: The memory manifold projected to the first two principal components (PCs) associated with the vector fields. Right panel: Example drift-diffusion trajectories along the memory manifold starting at $\theta = 112.5°$. (**B, C**) Velocity (**B**) and noise coefficients (**C**) corresponding to drift and diffusion processes. Different gray scales represent different heterogeneity degrees in the sensory module, $\alpha$, in *Figure 3B*. The velocity with which the remembered orientation drifts to the obliques in a noise-free network (**B**). A larger noise coefficient around the obliques overcomes the underlying drift dynamics and causes the standard deviation pattern to reach its maxima at the obliques (**C**). (**D**) Equivalent one-dimensional energy potential derived from the velocity in (**B**). (**E, F**) Example bias (**E**) and standard deviation (**F**) patterns at 4 s into the delay. The shaded areas mark the ±s.e.m. of 1000 realizations.

The online version of this article includes the following figure supplement(s) for figure 5:

**Figure supplement 1.** Comparison between bias and standard deviation (SD) patterns of the full network model (orangish) and low-dimensional projection (bluish curves).

**Figure supplement 2.** Standard deviation (SD) pattern remains consistent under different noise types.

external input overwhelms internal noise, which may not fully account for the variability observed during perception tasks (see Discussion). We obtained Fisher information measuring sensitivity at each orientation from the neural responses (see Methods). Opposite to the variance of errors, Fisher information is highest at cardinal orientations, while it decreases during the delay period (*Figure 4E*). Thus, the sensory-memory interacting network model that mechanistically embodies the extension of the Bayesian sensory model correctly reproduces the error patterns observed in delayed estimation tasks.

## Analysis of low-dimensional memory states

To further understand the mechanisms of generating the correct error patterns in sensory-memory interacting networks, we analyzed the network dynamics during the delay period. For this, we identified the low-dimensional manifold that has slow dynamics during the delay period, which corresponds to the memory states (*Figure 5A*). We projected the dynamics along this manifold to obtain the drift and diffusion terms (*Figure 5A–C*; *Figure 5—figure supplement 1*). The drift term shows similar patterns to cardinal repulsion (*Figure 5B and E*). Integrating this drift for orientation yields the energy function, which is minimum at the obliques (*Figure 5D*). This suggests that the network implements discrete attractor dynamics with attractors formed at the obliques. The diffusion term is also uneven – the noise amplitude is maximum at the obliques so that despite attraction toward them, the variance of errors can be maximum (*Figure 5C and F*). Note that while we use Poisson noise in all units to replicate neuronal spike variability, the pattern of noise coefficients remains unchanged even with constant Gaussian noise (*Figure 5—figure supplement 2*). This lower variance near cardinal orientations arises from more dispersed representations of stimuli, as the noise coefficient is inversely proportional to the distance between stimulus representations (*Equation 21*). Thus, the nonuniform characteristics

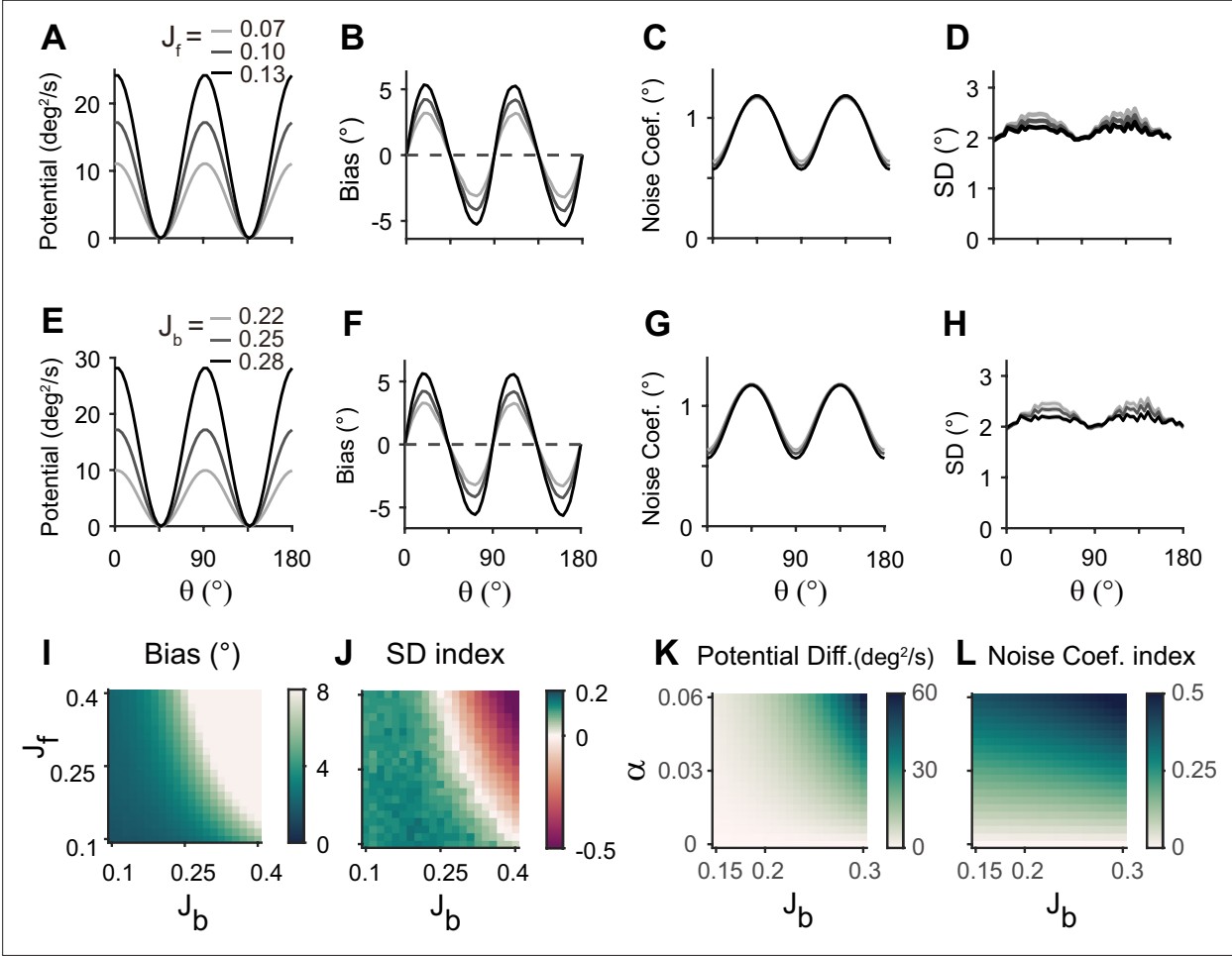

**Figure 6.** Error patterns and low-dimensional dynamics for different intermodal connectivity strengths. (**A–J**) Low-dimensional dynamics and error patterns with varying feedforward and feedback connection strengths, denoted by $J_f$ and $J_b$. (**K, L**) Potential differences and noise coefficient indices comparing low-dimensional dynamics at cardinal and oblique orientations for changing $J_b$ and heterogeneity degree, $\alpha$. Increasing both feedforward (**A–D**) and feedback (**E–H**) connection strengths deepens the potential difference (**A, E, K**) and increases the bias (**B, F**), similar to the effects of $\alpha$ increases in **Figure 5D and E**. In contrast, the profile of noise coefficients is less affected (**C, G, L**) and the SD pattern gets flattened with stronger drift (**D, H**). Bias and SD patterns depend on the product of feedforward and feedback connection strengths (**I, J**). Bias and SD are estimated at 4 s (**B, D, F, H**) or 1 s (**I,J**) into the delay and shaded areas mark the ±s.e.m. of 1000 realizations.

of both drift and diffusion processes stem from the heterogeneous connections within the sensory module and align with the solution identified in low-dimensional memory models (**Figure 1J–L**).

Next, we examined how heterogeneity of connectivity in the sensory module affects the dynamics along the memory states. The magnitude of heterogeneity is denoted as $\alpha$, and larger $\alpha$ represents a larger asymmetry of connectivity strengths at cardinal and oblique orientations (**Figure 3B**, left). When $\alpha$ increases, the asymmetry of drift and energy levels becomes more prominent, leading to a more rapid increase in bias (**Figure 5B, D, and E**). The diffusion term is also more asymmetric, compensating for stronger attraction to the obliques (**Figure 5C**). Thus, for larger $\alpha$, the variability of errors is still higher at the obliques (**Figure 5F**). Another important parameter influencing error patterns is the intermodal connectivity strengths (**Figure 6**). Similar to the effect of increasing $\alpha$, increases in feedforward or feedback strengths cause the energy levels to become more asymmetrical (**Figure 6A and E**), leading to a larger bias (**Figure 6B and F**). Conversely, the noise coefficient is less affected (**Figure 6C and G**), and the variance of errors decreases as the drift force becomes stronger (**Figure 6D and H**). Note that bias and variance patterns depend on the product of feedforward and feedback connections, denoted as $\gamma$, such that for a fixed $\gamma$, the error patterns remain similar (**Figure 6I and J**). In sum, the bias and variability of errors are determined by the degree of heterogeneity in the sensory module

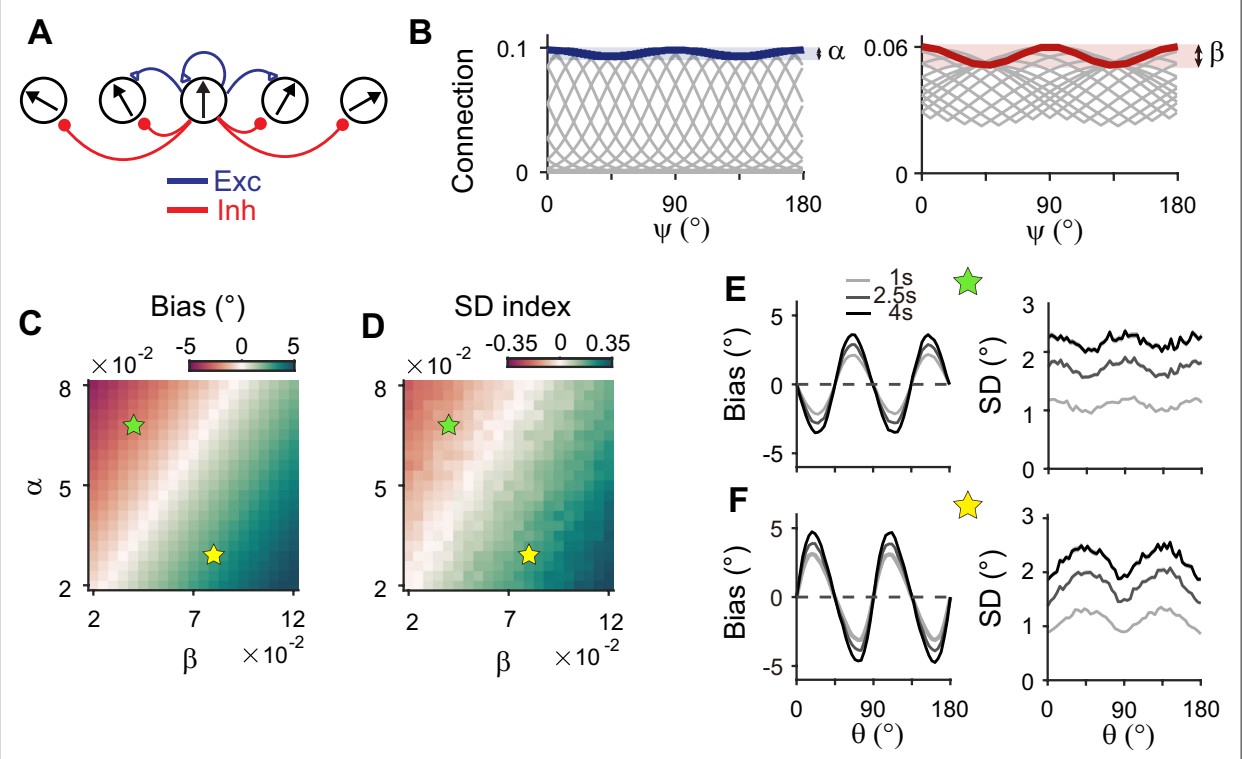

**Figure 7.** Stronger inhibitory synaptic modulation is required for correct error patterns. (**A**) Segregation of excitatory (blue) and inhibitory (red) synaptic pathways. (**B**) Example excitatory (left) and inhibitory (right) connectivity strengths of the sensory module. The heterogeneity degrees of excitatory and inhibitory connections are denoted by $\alpha$ and $\beta$, respectively. Unlike combined excitation and inhibition in *Figure 3B*, the connectivity strengths are maximal around cardinal orientations. (**C, D**) Bias with stimulus at 22.5° (**C**) and standard deviation (SD) index (**D**) estimated at 1 s into the delay for different values of $\alpha$ and $\beta$. SD index compares the SD at the cardinal and oblique orientations (Methods). (**E, F**) Example bias (left) and SD (right) patterns when excitatory modulation overwhelms inhibitory modulation ($\alpha = 0.068$, $\beta = 0.04$; **E**) and when inhibitory modulation is stronger ($\alpha = 0.03$, $\beta = 0.08$; **F**). In (**C**) and (**D**), green (yellow) pentagrams mark the parameters used in (**E**) and (**F**). Stronger inhibitory modulation is required for correct bias and variance patterns (**F**) and green regions in (**C** and **D**). In (**E**) and (**F**), shaded areas mark the ±s.e.m. of 1000 realizations.

The online version of this article includes the following figure supplement(s) for figure 7:

**Figure supplement 1.** Relationship between drift speed and memory loss in two-module (**A–C**) and one-module (**D–F**) networks.

**Figure supplement 2.** Error patterns in sensory networks with long intrinsic time constants.

($\alpha$) and intermodal connectivity strengths ($\gamma$) as both $\alpha$ and $\gamma$ affect the asymmetry of drift term similarly, while the asymmetry of diffusion term is more strongly influenced by $\alpha$ (*Figure 6K and L*).

## Importance of heterogeneously tuned inhibition

We showed that network models realizing sensory-memory interactions reproduce correct error patterns, where each module has a different connectivity structure. Previous work suggested that such a heterogeneous connection of the sensory system may arise from experience-dependent synaptic modification (*Olshausen and Field, 1996*; *Zylberberg et al., 2011*). For example, typical Hebbian learning is thought to potentiate connectivity strengths between neurons whose preferred stimuli are more frequently encountered. For orientations, cardinal directions are predominant in natural scenes. Thus, if experience-dependent learning occurs mainly at the excitatory synapses, the excitatory connections near cardinal orientations become stronger in the sensory module. This is opposite to the previously discussed case where the sensory module has the strongest connection at the obliques. With the strongest excitatory connections at cardinal orientations, the error patterns are reversed, resulting in cardinal attraction instead of repulsion, and the lowest variance occurs at the obliques.

Inhibitory synaptic connections can also be modified through learning (*Vogels et al., 2013*; *Khan et al., 2018*; *Larisch et al., 2021*). Here, we considered that experience-dependent learning exists in both excitatory and inhibitory pathways and similarly shapes their connectivity (*Figure 7A*). We

assumed that excitatory and inhibitory connections are segregated and stronger near cardinal orientations (*Figure 7B*). We modulated the heterogeneity degree of both excitatory and inhibitory connections, denoted as $\alpha$ and $\beta$, respectively (*Figure 7B–D*). The ratio between $\alpha$ and $\beta$ determines the direction and magnitude of bias and variance patterns (*Figure 7C and D*). For relatively larger $\alpha$, the network shows cardinal attraction and minimum variance of errors at the obliques (*Figure 7E*). Reversely, for relatively larger $\beta$ with stronger modulation in inhibitory connections, the network reproduced cardinal repulsion and minimum variance of errors at cardinal orientations, consistent with experiments (*Figure 7F*). With a larger difference between $\alpha$ and $\beta$, such patterns of bias and variance are potentiated and minimum Fisher information across orientations decreases, corresponding to memory loss (*Figure 7C and D*; *Figure 7—figure supplement 1*). Thus, this emphasizes the important role of heterogeneously tuned inhibition in shaping the sensory response for higher precision at cardinal orientations and enabling the sensory-memory interacting network to generate correct error patterns.

## Comparison to alternative circuit structures

So far, we have shown the sufficiency of sensory-memory interacting networks with different connectivity structures featuring heterogeneous-homogeneous recurrent connections within each module. Here, we explore whether such architecture is necessary by comparing its performance with alternative circuit structures for sensory-memory interactions. One candidate mechanism involves having the heterogeneous sensory network maintain memory with a long intrinsic time constant, similar to having autapses (*Seung et al., 2000*). However, this model fails to replicate the evolution of error patterns during the delay period as a long intrinsic time constant slows down the overall dynamics, thus hindering the evolution of error patterns (*Figure 7—figure supplement 2*). Alternatively, we focused on a two-module network with variations in connectivity structure. We assumed that sensory

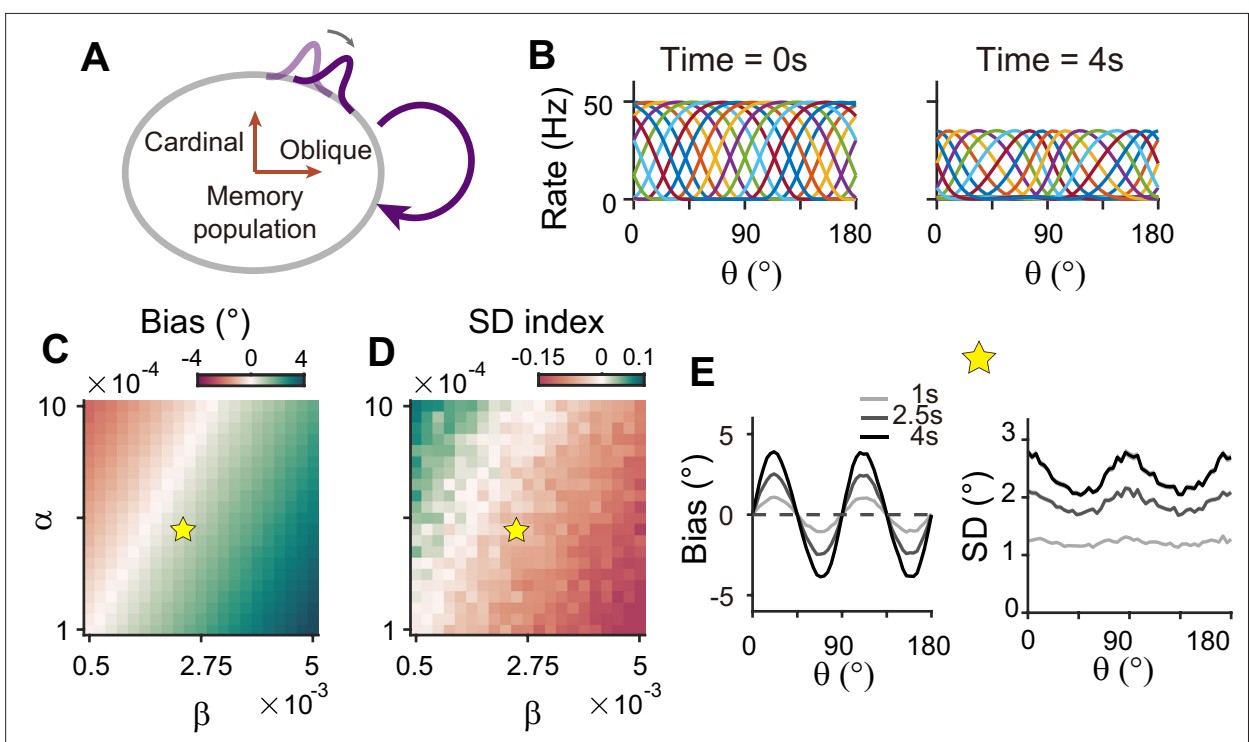

**Figure 8.** Network model with memory module only cannot reproduce correct error patterns. (**A**) Schematics of one-module network with heterogeneous and strong recurrent connections that enable both efficient coding and memory maintenance. (**B**) Example tuning curves at the end of the stimulus epoch (left) and at 4 s into the delay epoch (right). (**C, D**) Bias with stimulus at 22.5° (**C**) and standard deviation (SD) index (**D**) estimated at 1 s into the delay for different heterogeneity degrees of excitatory and inhibitory connections, denoted by $\alpha$ and $\beta$. For the parameters that generate reasonable bias patterns, the SD index is always negative, which indicates that the SD pattern is inconsistent with experimental findings. (**E**) Bias (left), and SD (right) patterns in the delay. While the bias pattern is correct, the SD reaches maxima around cardinal orientations, unlike the experiments. In (**C**) and (**D**), the yellow pentagram marks the parameters used in (**E**).

and memory modules still serve their distinctive functions, namely, sensory encoding and memory maintenance, with weak/strong recurrent connections in sensory/memory modules. On the other hand, the heterogeneity of connections in other circuits might differ as homogeneous-homogeneous, homogeneous-heterogeneous, and heterogeneous-heterogeneous connections for sensory-memory modules.

Circuits with homogeneous connections in both sensory and memory modules are similar to previous continuous attractor models for working memory, such that the energy landscape and noise amplitude are uniform for all orientations (*Figure 1D–F*). Such architecture is not suitable as it generates no bias in errors and flat variance patterns. This leaves the latter two types of configurations, which require heterogeneous connections within the memory module. With a strong recurrent connection within the memory module, its heterogeneous activity pattern dominates overall activities in sensory-memory interacting networks, which makes it analogous to an isolated memory module. Thus, we examined the property of the memory module alone, which can maintain memory while generating heterogeneous responses without connection to the sensory module (*Figure 8*).

To generate the correct bias pattern, we assumed that excitatory and inhibitory pathways in the memory module are stronger near cardinal orientations, as we previously considered for the sensory module in the sensory-memory interacting network (*Figure 8A and B*). However, memory circuits with heterogeneous connections have problems in maintaining the information and reproducing correct error patterns (*Figure 8C–E*). First, memory circuits alone require fine-tuning of heterogeneity whose range generating a moderate drift speed is at least one order of magnitude smaller than that of the two-module network (*Figure 8C and D*). Deviation from this range results in a fast drift toward oblique orientations, leading to rapid loss of information during the delay period (*Figure 7—figure supplement 1*). Second, despite the correct bias direction, the variance pattern is reversed such that the variance of errors is minimal at the oblique orientations (*Figure 8E*). Varying the heterogeneity in excitatory and inhibitory connections shows that such rapid drift and reversed error patterns are prevalent across different parameters (*Figure 8C and D*).

To understand why a heterogeneous memory circuit alone fails to reproduce correct error patterns, we compared its low-dimensional dynamics along the memory states to that of the sensory-memory interacting networks. For the network with a similar range of bias and variance on average, we compared their energy landscape and noise amplitude, which vary similarly in both networks with minimum energy level and maximum noise at the oblique orientations (*Figure 9A–F*). However, the energy difference between cardinal and oblique orientations in a single memory circuit model is bigger than that in a sensory-memory interacting network (*Figure 9C*, left in *Figure 9G, H*). In contrast, the difference in noise amplitude is smaller (*Figure 9D–F*, right in *Figure 9G, H*). The attraction at the obliques is much stronger, leading to the correct bias patterns, but too rapid an increase. Also, smaller differences in noise amplitude cannot overcome strong drift dynamics, leading to the minimum variance of errors at the obliques and reversed variance patterns. Even for different types or levels of noise, such as Gaussian noise with varying amplitude, distinctive error patterns in one-module and two-module networks are maintained (*Figure 9—figure supplement 1*).

For an intuitive understanding of how connectivity heterogeneity affects the degrees of asymmetry in drift and diffusion differently in one-module and two-module networks, consider a simple case where only the excitatory connection exhibits heterogeneity, the degree of which is denoted by $\alpha$. For memory maintenance, the overall recurrent connections need to be strong enough to overcome intrinsic decay, simplified to $w=1$. In the one-module network, $\alpha$ in the memory module causes deviations from perfect tuning, creating potential differences at cardinal and oblique orientations as $1\pm\alpha$. In the two-module network, with $w=1$ fulfilled by the memory module, $\alpha$ in the sensory module acts as a perturbation. The effect of $\alpha$ is modulated by the intermodal connectivity strengths, denoted by $\gamma$, and potential differences at cardinal and oblique orientations can be represented as $1\pm\gamma\alpha$. Thus, while a relatively large $\alpha$ leads to too fast drift in the one-module network, the drift speed in the two-module network could remain modest with small $\gamma<1$. Conversely, even with small $\gamma$, the asymmetry of noise coefficients can be large enough to produce correct variance patterns because the noise coefficient is more strongly influenced by $\alpha$ in the two-module network (*Figure 6*). In sum, compared to a heterogeneous memory circuit alone, interactions between heterogeneous sensory and homogeneous memory modules are advantageous due to an additional degree of freedom, intermodal

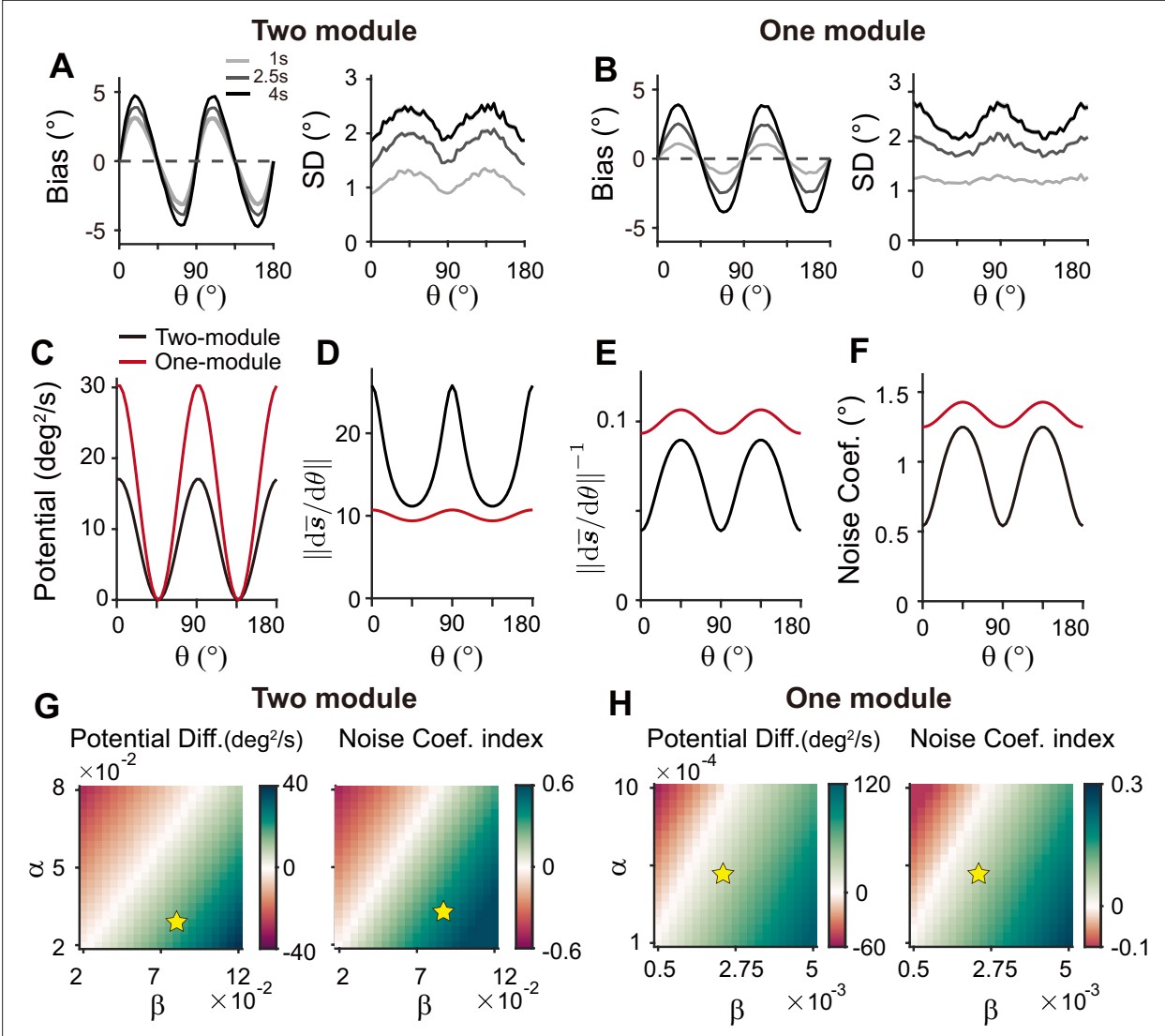

**Figure 9.** Comparison of low-dimensional dynamics between two-module and one-module network models. (**A, B**) Bias and standard deviation (SD) patterns of two-module (**A**) and one-module (**B**) networks, adapted from *Figure 7F* and *Figure 8E*, respectively. The averages of bias and SD over different $\theta$ at 4 s into the delay are similar in the two networks. (**C–F**) Low-dimensional dynamics of two-module (black) and one-module (red) networks. In both networks, the energy potential (**C**), the distance between stimulus representation, $\|\ \overline{\boldsymbol{s}}'\ (\theta)\ \|$ and its inverse determining noise coefficients (**D, E**; *Equation 21*), and the noise coefficients (**F**) exhibit similar profiles. However, the two-module network has a shallower potential (**C**) but larger heterogeneity in $\|\ \overline{\boldsymbol{s}}'\ (\theta)\ \|$ and the noise coefficient profile (**D–F**). These differences make it possible for the SD to become smaller around cardinal orientations in the two-module network (right in **A**), while drift dynamics overwhelm and the SD pattern is opposite to that of the noise coefficient in the one-module network (right in **B**). (**G, H**) Potential difference (left) and index of noise coefficients (right) comparing low-dimensional dynamics at the cardinal and oblique orientations in two-module (**G**) and one-module (**H**) networks. The two-module network shows a smaller potential difference and more heterogeneous noise coefficients over a broad range of heterogeneity (see the color bars in **G** and **H**).

The online version of this article includes the following figure supplement(s) for figure 9:

**Figure supplement 1.** Error patterns remain unchanged under different levels of noise.

connectivity strengths, which allows better control of energy and noise difference at cardinal and oblique orientations.

## Discussion

While higher association areas have long been considered as a locus of working memory (*Roussy et al., 2021*; *Mejías and Wang, 2022*), recent human studies found memory signals in early sensory

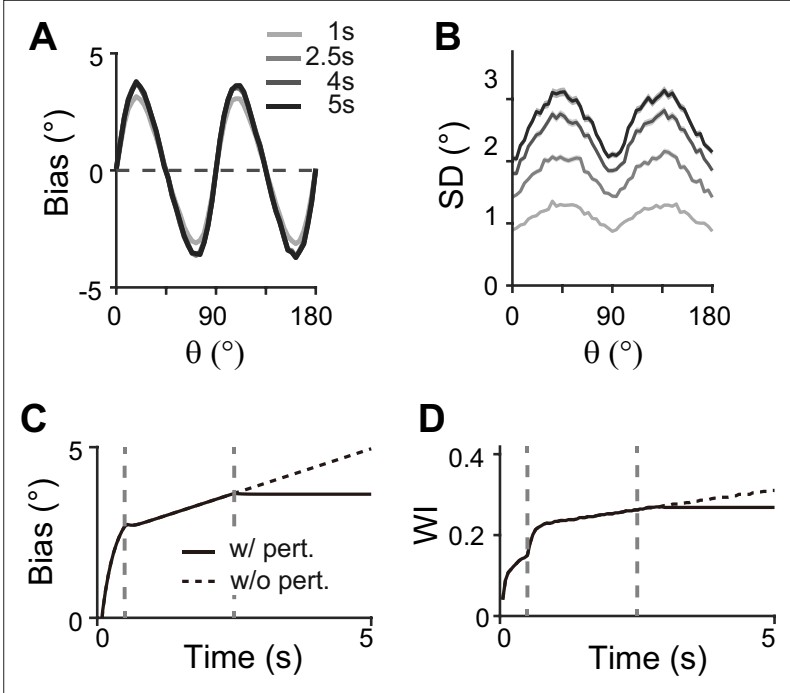

**Figure 10.** Effect of perturbations in sensory-memory interaction on error patterns. (**A, B**) Example bias (**A**) and standard deviation (**B**) patterns when we assumed that transcranial magnetic stimulation (TMS) is applied to interrupt the feedforward signal from 2.5 s into the delay. Shaded areas mark the ±s.e.m. of 1000 realizations. (**C, D**) Evolution of bias with example cue orientation at $\theta = 18°$ (**C**) and the tuning width indices in the memory network (WI; **C**) representing the asymmetry of tuning widths at cardinal and oblique orientations (Methods). Two vertical dashed lines mark the end of the stimulus epoch and the beginning of TMS disruption, respectively. Solid and dashed curves correspond to with and without perturbations, respectively. Both bias (**C**) and WI (**D**) stop increasing when TMS is on (**C, D**).

areas, prompting a re-evaluation of their role in working memory (*Xu, 2020*; *Adam et al., 2022*). Our work extends the traditional memory models (*Wang, 2001*; *Khona and Fiete, 2022*) with novel insights into the significance of stimulus-specific sensory areas. We showed how sensory-memory interactions can elucidate changes in the internal representation of orientation stimuli and their behavioral readout during memory tasks. The observed error patterns suggest that the network meets two demands simultaneously: efficient encoding that reflects natural statistics and memory maintenance for successful retrieval of stimuli after a delay. Achieving both demands for orientation stimuli conflicts in a one-module network. Efficient encoding necessitates asymmetrical connections, resulting in inconsistent bias and variance patterns and overly rapid drift in the one-module network unless fine-tuned. In contrast, connecting sensory and memory modules can generate error patterns correctly and with less need for fine-tuning heterogeneity for slow drift. Efficient coding of natural statistics in the sensory module underscores the role of inhibitory plasticity. Low-dimensional projection onto memory states reveals that drift and diffusion processes governing working memory dynamics closely resemble the bias and variance patterns derived under Bayesian sensory models. It also elucidates how the magnitudes of bias and variance change depending on the heterogeneity of sensory connections and intermodal connectivity strengths.

Our model makes testable predictions to differentiate two-module and one-module networks using perturbation, such as transcranial magnetic stimulation (TMS). Many studies have found that during the delay period, TMS can intervene with the feedforward signal from sensory areas through which working memory is consolidated (*van de Ven et al., 2012*) (but see *Adam et al., 2022*, for mixed effects of TMS and related debate). Under such perturbations, the ability to maintain information in the memory module will not be affected due to strong recurrent connections in both two-module and one-module networks. However, we expect different effects on bias patterns — in the two-module network, the bias will stop systematically drifting toward the obliques, reducing systematic repulsion

(*Figure 10*). This accompanies the nonincreasing heterogeneity of tuning curves after the disruption, marked by their tuning width indices (see Methods). In contrast, in the one-module network, perturbation does not incur changes in error patterns as memory activities are less dependent on the sensory module during the delay period. Thus, perturbation studies can be used to reveal the role of the sensory module in shaping the error patterns during working memory. Note that our model cannot predict the effects of distractors during working memory, as such effects do not experimentally lead to changes in error patterns (*Rademaker et al., 2019*). The effect of distractors and direct intervention in the intermodule connections may differ due to potential differences in the encoding of distractors compared to task-relevant stimuli. More advanced models are required to comprehensively understand the influence of distractors and the processing of ongoing visual stimuli or the storage of multiple stimuli.

Our work suggests biologically plausible network mechanisms for the previously postulated efficient coding and Bayesian inference principles, relating network connectivity to tuning properties and error patterns. Previous normative explanations for systematic bias observed in perception tasks also suggested possible neural substrates for efficient coding, such as asymmetrical gain, width, or density of tuning curves across stimulus features (*Ganguli and Simoncelli, 2014*; *Wei and Stocker, 2015*). Our work narrowed the mechanism to denser and narrower tuning curves at cardinal orientations, consistent with neurophysiological recordings in the visual cortex (*Li et al., 2003*; *Kreile et al., 2011*; *Shen et al., 2014*). We implemented a population vector decoder reflecting neuronal preferred orientations, which approximates Bayesian optimal readout (*Fischer, 2010*). Compared to a previous work adapting efficient coding theories with static tuning curves to account for error patterns in working memory tasks (*Taylor and Bays, 2018*), our extension to memory processes demonstrated how neural activities and behavior readout change dynamically during the delay period. Notably, recent work combined dynamic change of signal amplitude with static tuning curves to capture different time courses of estimation precision during sensory encoding and memory maintenance (*Tomić and Bays, 2023*). Our network models embody such phenomenological models as the networks exhibit changes in overall firing rates after the stimulus offset.

Like our study, a few recent studies have employed attractor dynamics to explain dynamic error patterns observed for visual color memory (*Panichello et al., 2019*; *Pollock and Jazayeri, 2020*; *Eissa and Kilpatrick, 2023*). Behavior studies showed attractive bias and minimum variance around the prevalent colors, which one-module discrete attractor models could reproduce. However, these models cannot be generalized to other visual stimuli, such as orientations, spatial locations, or directions, of which the responses show repulsive bias away from the common stimuli (*Wei and Stocker, 2017*). Also, a one-module network storing color memory requires fine-tuned heterogeneity for moderate drift speed. While the desired low-dimensional manifold and drift dynamics can be engineered in the one-module network (*Pollock and Jazayeri, 2020*), its biological mechanism needs further investigation. The two-module network considered in our study also requires fine-tuning of homogeneity in the memory module and heterogeneity in the sensory module. However, the condition of asymmetrical connections in the sensory module is less stringent as they have a weaker influence on the entire dynamics than those in the memory module. Fine-tuning of homogeneous connections in the memory module can be mediated through activity-dependent plasticity, such as short-term facilitation (*Itskov et al., 2011*; *Hansel and Mato, 2013*; *Seeholzer et al., 2019*) or long-term plasticity (*Renart et al., 2003*; *Gu and Lim, 2022*). Also, recent work showed that continuous attractors formed under unstructured, heterogeneous connections are robust against synaptic perturbations (*Darshan and Rivkind, 2022*). Thus, the two-module networks can control the drift speed better with possible additional mechanisms that promote homogeneous memory states. It needs further exploration whether they can be generalized to other stimuli like color, possibly involving additional categorical structures (*Hardman et al., 2017*; *Pratte et al., 2017*).

Our current study is limited to the dynamic evolution of memory representation for a single orientation stimulus and its associated error patterns, which does not capture nuanced error patterns in broader experimental settings (*Hahn and Wei, 2024*). For instance, while shorter stimulus presentations with no explicit delay led to larger biases experimentally, our current model, which starts activities from a flat baseline, shows an increase in bias throughout the stimulus presentation (*de Gardelle et al., 2010*). Additionally, the error variance during stimulus presentation is almost negligible compared to that during the delay period, as the external input overwhelms the internal noise.

These mismatches during stimulus presentation have minimal impact on activities during the delay period when the internal dynamics dominate. Nonetheless, the model needs further refinement to accurately reproduce activities during stimulus presentation, possibly by incorporating more biologically plausible baseline activities. Also, a recent Bayesian perception model suggested different types of noise like external noise or variations in loss functions that adjust tolerance to small errors may help explain various error patterns observed across different modalities (*Hahn and Wei, 2024*). Even for memories involving multiple items, noise can be critical in determining error patterns, as encoding more items might cause higher noise for each individual item (*Chunharas et al., 2022*).

The modularity structure in the brain is thought to be advantageous for fast adaptation to changing environments (*Simon, 1995*; *Cole et al., 2013*; *Frankland and Greene, 2020*). Recent works showed that recurrent neural networks trained for multiple cognitive tasks form clustered neural activities and modular dynamic motifs to repurpose shared functions for flexible computation (*Yang et al., 2019*; *Driscoll et al., 2022*). Resonant with these computational findings, an fMRI study showed that shared representation across distinct visual stimuli emerges during the delay period (*Kwak and Curtis, 2022*). Although our work focuses on a single task, it highlights the necessity of having dedicated sensory and memory modules, and a memory module with ring geometry can be repurposed for various visual stimuli such as motion, spatial location, and color. It is reminiscent of the flexible working memory model, which proposes connections between multiple sensory modules and a control module (*Bouchacourt and Buschman, 2019*). However, a key distinction lies in the role of the control module. Unlike the flexible working memory model that loses memory without sensory-control interactions, our work suggests that the memory module can independently maintain memory, while interaction with the sensory module continuously shapes the internal representation, potentially consolidating prior beliefs regarding natural statistics. The sensory-memory interaction and network architecture derived from dynamic changes of single stimulus representation can be a cornerstone for future studies in more complex conditions, such as under the stream of visual inputs (*Xu, 2020*; *Adam et al., 2022*) or with high or noisy memory loads (*Bays et al., 2022*).

## Methods
### Low-dimensional attractor models
To illustrate error patterns in different low-dimensional attractor models shown in *Figure 1*, we considered a one-dimensional stochastic differential equation given as

$$\mathrm{d}\theta_t = \mu\left(\theta_t\right)\mathrm{d}t + \sigma\left(\theta_t\right)\mathrm{d}W_t, \tag{1}$$

where $\theta_t$ and $W_t$ are orientation and standard Brownian motion at time $t$, respectively. We assumed that the drift and noise coefficients $\mu$ and $\sigma$ only depend on $\theta_t$, where $\sigma = \sqrt{2\mathfrak{D}}$ with diffusion coefficient $\mathfrak{D}$.

For continuous attractor models in *Figure 1D–F*, $\mu$ and $\sigma$ were set to be constant as $\mu = 0$ and $\sigma = 2°$. For discrete attractor models in *Figure 1G–L*, we assumed that the energy function $U\left(\theta_t\right)$ is proportional to $\cos\left(4\theta_t\right)$ (*Figure 1G and J*) so that the drift term $\mu\left(\theta_t\right) = \sin\left(4\theta_t\right)$ with $\mu\left(\theta_t\right) = -\frac{dU}{d\theta_t}$. In these attractor models, the constant noise in *Figure 1G–I* is $\sigma = 2°$ and the nonuniform noise in *Figure 1J–L* is $\sigma = 2°\left(1 - \cos\left(4\theta_t\right)\right)$. The biases and standard deviation (SD) of errors were plotted at $T$=1, 2, and 3 with 50,000 iterations. For the numerical simulation, d$t$ =0.01.

### Bayesian sensory models and extension
In *Figure 2*, we first constructed the sensory inference process, which receives orientation input $\theta$, forms a corresponding noisy sensory representation $m$ given $\theta$, and then infers $\hat{\theta}$ as an estimate of the input orientation from the encoded representation $m$. This inference is made in a Bayesian manner based on likelihood function $p\left(m|\theta\right)$ and orientation prior $q\left(\theta\right)$.

To construct $p\left(m|\theta\right)$, we followed the procedure given in *Wei and Stocker, 2015*, and the summary is as follows. We started from the sensory space of $\tilde{\theta}$ where both discriminability and Fisher information $J\left(\tilde{\theta}\right)$ are uniform, and all likelihood functions $p\left(m|\tilde{\theta}\right)$ are homogeneous von Mises functions. And since $J\left(\theta\right) \propto \left(q\left(\theta\right)\right)^2$ under the efficient coding condition, the sensory space of $\tilde{\theta}$ and the stimulus space of $\theta$ can be mapped by forward and backward mappings $F\left(\theta\right)$ and $F^{-1}(\tilde{\theta})$, where $F\left(\theta\right)$ is the

cumulative distribution function of prior $q(\theta)$. Thus, likelihood functions $p(m|\theta)$ can be obtained by taking homogeneous von Mises likelihoods in the sensory space and transforming them back to the stimulus space using $F^{-1}$. To sum up the upper half of the procedural diagram in *Figure 2A*, the sensory module receives $\theta$, encodes it in $m$ following $p(m|\theta)$, and decodes $\hat{\theta}$ using likelihood functions and prior $q(\theta)$.

As an extension to include a memory process, the decoded $\hat{\theta}$ is passed on to the memory module, where $\hat{\theta}$ is maintained with the addition of memory noise $\xi$. The output of the memory module, $\hat{\theta} + \xi$, is fed back to the sensory module as the new input. This completes one iteration of sensory-memory interaction. The whole process is then repeated recursively, resulting in increased biases and standard deviations in the $\theta$ statistics at subsequent iterations (call them $\theta_i$ for the input of iteration $i$).

For *Figure 2B and C*, we set the von Mises sensory-space likelihoods to be $p(m|\tilde{\theta}) \propto \exp(\kappa_m \cos(m - \tilde{\theta}))$, with $\kappa_m = 250$. These likelihood functions are transformed by $F^{-1}(\tilde{\theta}) = \{\int q(\theta)\}^{-1}$, where $q(\theta) = 3 + \cos(4\theta)$. Each internal representation $m$ is sampled from $p(m|\theta)$, after which $\hat{\theta}$ is estimated as the mean of the posterior $p(\theta|m)q(\theta)$. With the parameters chosen above, the inferred samples of $\hat{\theta}$ after the first sensory iteration have a circular standard deviation of $\sigma_\theta \approx 1.3°$ at cardinal orientations. To have comparable memory and sensory noise levels, we set the memory noise as $\xi \sim \mathfrak{N}(0, (1.3°)^2)$ which is added on top of the sensory outputs. Thus, the memory outputs of the first iteration $\theta_1 = \hat{\theta}_1 + \xi$ have a standard deviation of 1.84° at the cardinals. The first three iterations' memory output statistics are plotted in *Figure 2C*, i.e., bias($\theta_1$), bias($\theta_2$), bias($\theta_3$), and SD($\theta_1$), SD($\theta_2$), SD($\theta_3$). The statistics were computed from 10,000 iterations of the simulation. The magnitude of biases and standard deviations vary for different sensory or memory noise levels, while the overall patterns and the increasing temporal trend are unchanged (not shown).

## Firing rate models

For network models, we considered sensory circuits with heterogeneous connections (*Figure 3*), memory circuits with homogeneous connections (*Figure 3*) and heterogeneous connections (*Figures 8 and 9*), and sensory-memory interacting circuits (*Figures 4–7, 9, and 10*). In all cases, the activities of neurons are described by their firing rates and synaptic states, denoted by $r$ and $s$. For columnar structure encoding orientation stimuli, we indexed the neurons by uniformly assigning them indices $\psi_i = \frac{(i-1)}{N} \times 180°$ for $i$ from 1 to $N$, where $N$ is the number of neurons in each population. For sensory or memory networks alone, the dynamics of neuron $i$ are described by the following equations:

$$
\begin{aligned}
r_k^i &= f_k\left(\sum_j W_k^{ij} s_k^j + I_{\text{ext},k}^i\right) \\
\tau \dot{s}_k^i &= -s_k^i + r_k^i + \xi_k^i
\end{aligned}
\tag{2}
$$

where the superscripts $i$ and $j$ are the neuronal indices, and the subscript $k$ is either s or m, representing sensory or memory circuits. For the sensory-memory interacting network, the dynamics are given as

$$
\begin{aligned}
\boldsymbol{r}_{\text{s}} &= f_{\text{s}}\left(\boldsymbol{W}_{\text{s}}\boldsymbol{s}_{\text{s}} + \boldsymbol{W}_{\text{b}}\boldsymbol{s}_{\text{b}} + \boldsymbol{I}_{\text{ext,s}}\right) \\
\boldsymbol{r}_{\text{m}} &= f_{\text{m}}\left(\boldsymbol{W}_m\boldsymbol{s}_{\text{m}} + \boldsymbol{W}_{\text{f}}\boldsymbol{s}_{\text{f}} + \boldsymbol{I}_{\text{ext,m}}\right) \\
\tau \dot{\boldsymbol{s}}_k &= -\boldsymbol{s}_k + \boldsymbol{r}_{\text{s}} + \boldsymbol{\xi}_k, \quad \text{for } k = \text{s or f} \\
\tau \dot{\boldsymbol{s}}_l &= -\boldsymbol{s}_l + \boldsymbol{r}_{\text{m}} + \boldsymbol{\xi}_l, \quad \text{for } l = \text{m or b}
\end{aligned}
\tag{3}
$$

where activities and synaptic inputs are represented in the vector and matrix multiplication form, shown in bold cases. The additional subscripts f and b represent feedforward and backward connections between sensory and memory modules.

In both *Equations 2 and 3*, $s(t)$ is the low pass filtered $r(t)$ with synaptic time constant $\tau$ and with the addition of $\xi$ approximating Poisson noise. We modeled $\xi$ as the Gaussian process with covariance $\langle \xi^i(t)\xi^j(t')\rangle = r^i(t)\delta_{ij}\delta(t - t')$, following *Burak and Fiete, 2012*. We assumed that the rate dynamics are relatively fast such that $r(t)$ equals the input current-output rate transfer function $f$.

The input current is the sum of external input $I_{\text{ext}}$ and the synaptic currents from other neurons in the network, which are the postsynaptic states $s^j$ weighted by synaptic strengths $W^{ij}$. The transfer function $f$ has the Naka-Rushton form (**Wilson, 1999**) given as

$$f(x) = f_{max} \frac{(x - T)^q}{w^q + (x - T)^q} \cdot [x - T]_+ ,\qquad(4)$$

where $[\cdot]_+$ denotes the linear rectification function. The transfer functions differ in the sensory and memory modules, denoted as $f_{\text{s}}$ and $f_{\text{m}}$, respectively.

## Synaptic inputs in network models

Note that for all network models, we only considered excitatory neurons under the assumption that the inhibitory synaptic pathways have relatively fast dynamics. Thus, recurrent connectivity strengths, $W_{\text{s}}$ and $W_{\text{m}}$, within sensory and memory modules, reflect summed excitation and inhibition, and thus, can have either positive or negative signs. On the other hand, we assumed that intermodal interactions, $W_{\text{f}}$ and $W_{\text{b}}$, are dominantly excitatory and, thus, can be only positive.

All $W$'s can be defined using neuronal indices of post- and presynaptic neurons as

$$W^{ij} = \frac{1}{N} J\left(\psi_i, \psi_j\right) .\qquad(5)$$

For $W_{\text{s}}$ without segregating excitation and inhibition in **Figures 3–6**, $N$ is the population size of sensory module, $N_{\text{s}}$, and $J_{\text{s}}$ is the sum of a constant global inhibition and a short-range excitatory connection as

$$J_{\text{s}}\left(\psi_i, \psi_j\right) = -J_{\text{I,s}} + J_{\text{E,s}}\left(1 - \alpha\cos4\psi_i\right) e^{-\frac{\left(\psi_i - \psi_j\right)^2}{\lambda_{\text{E,s}}^2}} ,\qquad(6)$$

where $\alpha > 0$ represents the heterogeneity degree of excitatory connectivity, and $\lambda_{\text{E}}$ is the width of local excitatory connections.

When we segregated excitation and inhibition and considered the heterogeneity of inhibitory connection in **Figures 7–10**, **Equation 6** is replaced with

$$J_{\text{s}}\left(\psi_i, \psi_j\right) = -J_{\text{I,s}}\left(1 + \beta\cos4\psi_i\right) e^{-\frac{\left(\psi_i - \psi_j\right)^2}{\lambda_{\text{I,s}}^2}} + J_{\text{E,s}}\left(1 + \alpha\cos4\psi_i\right) e^{-\frac{\left(\psi_i - \psi_j\right)^2}{\lambda_{\text{E,s}}^2}} ,\qquad(7)$$

where $\beta > 0$ is the degree of heterogeneity of inhibitory connections. Note the signs of modulation change in **Equations 6 and 7** such that when only excitation is modulated in **Equation 6**, the connectivity strengths near the obliques are strong. In contrast, when excitation and inhibition are both modulated in **Equation 7**, the connectivity strengths near cardinal orientations are strong.

For the memory module, $N$ is the population size of the memory module, $N_{\text{m}}$ in **Equation 5**. Without heterogeneity in **Figures 3–7 and 10**, $J_{\text{m}}$ is defined as

$$J_{\text{m}}\left(\psi_i, \psi_j\right) = -J_{\text{I,m}} e^{-\frac{\left(\psi_i - \psi_j\right)^2}{\lambda_{\text{I,m}}^2}} + J_{\text{E,m}} e^{-\frac{\left(\psi_i - \psi_j\right)^2}{\lambda_{\text{E,m}}^2}} .\qquad(8)$$

In contrast, for the one-module network model in **Figure 8**, the connectivity of the memory module is heterogeneous, as in the sensory module in **Equation 1**, and is defined as

$$J_{\text{m}}\left(\psi_i, \psi_j\right) = -J_{\text{I,m}}\left(1 + \beta\cos4\psi_i\right) e^{-\frac{\left(\psi_i - \psi_j\right)^2}{\lambda_{\text{I,m}}^2}} + J_{\text{E,m}}\left(1 + \alpha\cos4\psi_i\right) e^{-\frac{\left(\psi_i - \psi_j\right)^2}{\lambda_{\text{E,m}}^2}} .\qquad(9)$$

The feedforward and feedback connectivity are similarly defined as

$$W_f^{ij} = \frac{1}{N_s} J_f e^{-(\psi_{mi} - \psi_{sj})^2/\lambda_f^2}$$
$$W_b^{ij} = \frac{1}{N_m} J_b e^{-(\psi_{si} - \psi_{mj})^2/\lambda_b^2}. \tag{10}$$

Note the connectivity strength is normalized by the size of the presynaptic population so that the total synaptic current remains the same for different population sizes.

For the external inputs with orientation $\theta$, $I_{ext,s}$ in the sensory module is modeled as

$$I_{ext,s}^i(\theta) = C\left(1 - 2\varepsilon + 2\varepsilon e^{-(\psi_i - \theta)^2/\lambda_{ext,s}^2}\right), \tag{11}$$

where $\varepsilon \in (0, 0.5]$ determines the stimulus tuning of the input, $\lambda_{ext,s}$ determines the width, and $C$ describes the contrast (**Hansel and Sompolinsky, 1998**).

For the memory network not connected to the sensory module in **Figures 3 and 8**, we assumed stimulus-specific input as

$$I_{ext,m}^i(\theta) = \frac{1}{2}\left(\cos\left(2\left(\psi_i - \theta\right)\right) + 1\right) + I_{c,m}, \tag{12}$$

where $I_{c,m}$ is a constant background input. When the memory module receives the inputs from the sensory population in **Figures 4–7 and 10**, we assumed $I_{ext,m}^i(\theta)$ is constant as $I_{c,m}$.

## Analysis of network activities

We used population vector decoding to extract the internal representation of orientation and quantified how such representation deviated from the original stimulus. We also examined how tuning properties and Fisher information change during the delay period.

Note that while we indexed neurons uniformly with $\psi_i$ between 0° and 180°, the maximum of the tuning curve of neuron $\psi_i$ can change dynamically and differ from $\psi_i$. We defined the preferred feature (PF) of neuron $i$ as the maximum of its tuning curve when the tuning curve reaches a steady state in the presence of external input. For numerical estimation, we set the stimulus-present encoding epoch to 5 s to obtain the steady states of tuning curves. The tuning width is given as the full width at half maximum (FWHM) of the tuning curve. To estimate PF and FWHM, we did a cubic spline interpolation to increase the number of sample orientations to 1000. The tuning width index (WI) is given as

$$WI = \frac{FWHM\left(\psi = 45°\right) - FWHM\left(\psi = 0°\right)}{FWHM\left(\psi = 45°\right) + FWHM\left(\psi = 0°\right)}. \tag{13}$$

To estimate the internal representation of orientation in the network models, denoted as $\hat{\theta}$, we utilized the population vector decoder (**Georgopoulos et al., 1986**)

$$\hat{\theta}(t) = \frac{1}{2}\mathrm{Arg}\left(\sum_{j=1}^{N} \exp\left\{2\mathrm{i}r^j(t)\,\tilde{\theta}_j\right\} / \sum_{j=1}^{N} r^j\right), \tag{14}$$

where $N$ denotes the number of neurons and $\tilde{\theta}_j$ denotes the PF of neuron $j$. The orientation is always decoded from the memory network tuning curves $r_m(t)$ except for **Figure 10A**. The estimation bias $b(\theta, t) = E\left[\hat{\theta}(t)\right] - \theta$. Since the bias is typically small enough, we computed the estimation standard deviation (SD) as the SD of bias using linear statistics. The SD index is defined as

$$SD\ index = \frac{SD\left(\theta = 45°\right) - SD\left(\theta = 0°\right)}{SD\left(\theta = 45°\right) + SD\left(\theta = 0°\right)}. \tag{15}$$

The Fisher information (FI) is estimated by assuming that the probability density function $p\left(r \mid \theta\right)$ is Gaussian as

$$p\left(r_m^i \mid \theta\right) = \frac{1}{\sqrt{2\pi}\sigma_i(\theta)} e^{\frac{\left(r_m^i(\theta) - E\left[r_m^i(\theta)\right]\right)^2}{2\sigma_i^2(\theta)}}, \tag{16}$$

where $\sigma_i^2(\theta) = \mathrm{Var}\left(r_{\mathrm{m}}^i(\theta)\right)$ denotes the variance of the firing rate of memory neuron $i$. Thus, we can estimate the FI of memory neuron $i$ based on the empirical mean and variance of the firing rate at time $t$ as

$$\mathrm{FI}\left(\psi_i, t\right) = \frac{\left(\partial E\left[r_{\mathrm{m}}^i(\theta, t)\right]/\partial\theta\right)^2}{\sigma_i^2(\theta, t)}, \tag{17}$$

and the total FI is the summation of the FI of all memory neurons, given as $\mathrm{FI}(t) = \sum_i \mathrm{FI}(\psi_i, t)$.

## Drift and diffusivity in network models

Although the modulation breaks the continuity of the ring attractor and forms two discrete attractors at the obliques, there is still a one-dimensional trajectory $\bar{s}(\theta)$ to which the noise-free dynamics quickly converge. We can linearize the system in the vicinity of this trajectory if the noise is small (*Burak and Fiete, 2012*). Note that the dynamics of the synaptic variables in *Equation 3* can be put into the following form:

$$\tau\dot{s} = -s + \phi\left(Ws + h\right) + \xi, \tag{18}$$

and by linearizing around the stable trajectory $s = \bar{s}$, we get

$$\tau\dot{\delta s} = K\delta s + \xi, \tag{19}$$

where we have ignored the zeroth- and higher-order terms. The drift velocity $\mu(\theta)$ is estimated by projecting the noise-free dynamics along the normalized right eigenvector $u$ of $K$ with the largest real part of the eigenvalue

$$\mu(\theta) = \frac{1}{\tau \parallel \bar{s}'(\theta) \parallel} u^{\mathrm{T}}(\theta)\left[-\bar{s}(\theta) + \phi\left(W\bar{s}(\theta) + h(\theta)\right)\right]. \tag{20}$$

The coefficient of diffusion can be obtained in the same way

$$2\mathfrak{D}(\theta) = \frac{1}{\left(\tau \parallel \bar{s}'(\theta) \parallel\right)^2}\sum_i u_i^2(\theta)\,\phi_i\left(\sum_j W_{ij}\bar{s}_j(\theta) + h_i\right). \tag{21}$$

The noise coefficient is given as $\sigma = \sqrt{2\mathfrak{D}}$. Hence, we have reduced the high-dimensional dynamics to a simple one-dimensional stochastic differential equation as in *Equation 1* as

$$\mathrm{d}\theta = \mu(\theta)\,\mathrm{d}t + \sigma(\theta)\,\mathrm{d}W_t,$$

and the potential $U(\theta)$ is obtained by the relation $\frac{\mathrm{d}U}{\mathrm{d}\theta} = -\mu(\theta)$. To quantitatively measure the heterogeneity of noise coefficient across different orientations, we define the noise coefficient index as follows:

$$\mathrm{Noise\ Coef.index} = \frac{\sigma\left(\theta = 45°\right) - \sigma\left(\theta = 0°\right)}{\sigma\left(\theta = 45°\right) + \sigma\left(\theta = 0°\right)}. \tag{22}$$

## Network parameters and simulations

Unless otherwise specified, $N_{\mathrm{s}} = N_{\mathrm{m}} = 300$, $\tau = 10\,\mathrm{ms}$. The connectivity parameters are $J_{\mathrm{E,s}} = 0.6, J_{\mathrm{I,s}} = 0.35, J_{\mathrm{E,m}} = 1, J_{\mathrm{I,m}} = 0.17$, $J_{\mathrm{f}} = 0.1, J_{\mathrm{b}} = 0.25, \lambda_{\mathrm{E,s}} = 0.36\pi, \lambda_{\mathrm{I,s}} = 1.1\pi, \lambda_{\mathrm{E,m}} = 0.2\pi$, $\lambda_{\mathrm{I,m}} = 0.6\pi, \lambda_{\mathrm{f}} = \lambda_{\mathrm{b}} = 0.17\pi$. For the external input, we set $C = 4, \varepsilon = 0.2$, and $\lambda_{\mathrm{ext,s}} = 0.3\pi$. For the modulation of the sensory network, unless otherwise specified, we set $\alpha = 0.04$ when only the excitatory plasticity is considered, and $\alpha = 0.03, \beta = 0.08$ when the inhibitory plasticity is added. As for the modulation of the single-layer memory network, we set $\alpha = 5 \times 10^{-4}, \beta = 2.4 \times 10^{-3}$. For the transfer function, $f_{\max} = 100, T = 0.1, q = 2, w = 6$ for sensory $f_{\mathrm{s}}$, and $f_{\max} = 100, T = 0.1, q = 1.5, w = 6.6$ for memory $f_{\mathrm{m}}$.

We uniformly sampled 50 cue orientations in $[0°, 180°]$. The visual cue lasts for 0.5 s except for the estimation of the PFs. In the grid parameter search figures, the delay epochs last for 1 s. In *Figure 3*, we set $\alpha = 0.07$. In *Figure 5A*, the manifold corresponds to the synaptic variables at 4 s into the delay with $\alpha = 0.05$. We uniformly sampled 100 cue orientations for the manifold.

To compute the drift velocity and noise coefficient in *Figures 5, 6, and 9*, we use the stable trajectory $\bar{s}(\theta)$ at 1 s into the delay to ensure the fast transient dynamics induced by stimulus offset fully decays. The stable trajectory is parameterized by the 50 cue orientations to numerically compute $\bar{s}'(\theta)$.

All simulations of ordinary or stochastic differential equations of the network models were done using the Euler method with $dt = 1\text{ms}$. We checked that similar results hold for smaller $dt$. Example bias and standard deviation patterns were estimated from 1000 independent realizations. The Fisher information patterns were estimated from 3000 independent realizations. The grid search of maximum bias at $\theta = 22.5°$ and standard deviation index were computed from 3000 realizations.

All simulations were run in MATLAB. The code is available at GitHub (copy archived at *Yang, 2024*).

## Acknowledgements

We appreciate X Wei for sharing the code for Bayesian inference models. JY was supported by the NYU Shanghai Summer Undergraduate Research Program (SURP). SL received STI2030-Major Projects, No. 2021ZD0203700/2021ZD0203705. HZ and SL also acknowledge the support of the Shanghai Frontiers Science Center of Artificial Intelligence and Deep Learning and the NYU-ECNU Institute of Brain and Cognitive Science at NYU Shanghai.

## Additional information

### Funding

| Funder | Grant reference number | Author |
| --- | --- | --- |
| Ministry of Science and Technology of the People's Republic of China | STI2030-Major Projects No.2021ZD0203700 | Sukbin Lim |
| Ministry of Science and Technology of the People's Republic of China | 2021ZD0203705 | Sukbin Lim |
| NYU Shanghai | Summer Undergraduate Research Program (SURP) | Jun Yang |

The funders had no role in study design, data collection and interpretation, or the decision to submit the work for publication.

### Author contributions

Jun Yang, Hanqi Zhang, Conceptualization, Data curation, Software, Formal analysis, Investigation, Visualization, Writing – original draft, Writing – review and editing; Sukbin Lim, Conceptualization, Formal analysis, Supervision, Funding acquisition, Visualization, Writing – original draft, Project administration, Writing – review and editing

### Author ORCIDs

Jun Yang ⬤ https://orcid.org/0000-0002-2484-2494
Sukbin Lim ⬤ https://orcid.org/0000-0001-9936-5293

Reviewer #1 (Public review): https://doi.org/10.7554/eLife.95160.4.sa1
Reviewer #2 (Public review): https://doi.org/10.7554/eLife.95160.4.sa2
Reviewer #3 (Public review): https://doi.org/10.7554/eLife.95160.4.sa3
Author response https://doi.org/10.7554/eLife.95160.4.sa4

# Additional files

## Supplementary files
• MDAR checklist

## Data availability
The current manuscript is a computational study, so no data have been generated for this manuscript. The code is available at GitHub (copy archived at *Yang, 2024*).

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
