## [Editor Report · eLife assessment]

This **important** computational study provides new insights into how neural dynamics may lead to time-evolving behavioral errors as observed in certain working-memory tasks. By combining ideas from efficient coding and attractor neural networks, the authors construct a two-module network model to capture the sensory-memory interactions and the distributed nature of working memory representations. They provide **convincing** evidence supporting that their two-module network, although none of the alternative circuit structures they considered can account for error patterns reported in orientation-estimation tasks with delays.

---

## [Referee Report · Reviewer #1 (Public review)]

Summary:

Working memory is imperfect - memories accrue error over time and are biased towards certain identities. For example, previous work has shown memory for orientation is more accurate near the cardinal directions (i.e., variance in responses is smaller for horizontal and vertical stimuli) while being biased towards diagonal orientations (i.e., there is a repulsive bias away from horizontal and vertical stimuli). The magnitude of errors and biases increase the longer an item is held in working memory and when more items are held in working memory (i.e., working memory load is higher). Previous work has argued that biases and errors could be explained by increased perceptual acuity at cardinal directions. However, these models are constrained to sensory perception and do not explain how biases and errors increase over time in memory. The current manuscript builds on this work to show how a two-layer neural network could integrate errors and biases over a memory delay. In brief, the model includes a 'sensory' layer with heterogenous connections that lead to the repulsive bias and decreased error at the cardinal directions. This layer is then reciprocally connected with a classic ring attractor layer. Through their reciprocal interactions, the biases in the sensory layer are constantly integrated into the representation in memory. In this way, the model captures the distribution of biases and errors for different orientations that has been seen in behavior and their increasing magnitude with time. The authors compare the two-layer network to a simpler one-network model, showing that the one model network is harder to tune and shows an attractive bias for memories that have lower error (which is incompatible with empirical results).

Strengths:

The manuscript provides a nice review of the dynamics of items in working memory, showing how errors and biases differ across stimulus space. The two-layer neural network model is able to capture the behavioral effects as well as relate to neurophysiological observations that memory representations are distributed across sensory cortex and prefrontal cortex.

The authors use multiple approaches to understand how the network produces the observed results. For example, analyzing the dynamics of memories in the low-dimensional representational space of the networks provides the reader with an intuition for the observed effects.

As a point of comparison with the two-layer network, the authors construct a heterogenous one-layer network (analogous to a single memory network with embedded biases). They argue that such a network is incapable of capturing the observed behavioral effects but could potentially explain biases and noise levels in other sensory domains where attractive biases have lower errors (e.g., color).

The authors show how changes in the strength of Hebbian learning of excitatory and inhibitory synapses can change network behavior. This argues for relatively stronger learning in inhibitory synapses, an interesting prediction.

The manuscript is well-written. In particular, the figures are well done and nicely schematize the model and the results.

Weaknesses:

Despite its strengths, the manuscript does have some weaknesses. These weaknesses are adequately discussed in the manuscript and motivate future research.

One weakness is that the model is not directly fit to behavioral data, but rather compared to a schematic of behavioral data. As noted above, the model provides insight into the general phenomenon of biases in working memory. However, because the models are not fit directly to data, they may miss some aspects of the data.

In addition, directly fitting the models to behavioral data could allow for a broader exploration of parameter space for both the one-layer and two-layer models (and their alternatives). Such an approach would provide stronger support for the papers claims (such as "....these evolving errors...require network interaction between two distinct modules."). That being said, the manuscript does explore several alternative models and also acknowledges the limitation of not directly fitting behavior, due to difficulties in fitting complex neural network models to data.

One important behavioral observation is that both diffusive noise and biases increase with the number of items in working memory. The current model does not capture these effects and it isn't clear how the model architecture could be extended to capture these effects. That being said, the authors note this limitation in the Discussion and present it as a future direction.

Overall:

Overall, the manuscript was successful in building a model that captured the biases and noise observed in working memory. This work complements previous studies that have viewed these effects through the lens of optimal coding, extending these models to explain the effects of time in memory. In addition, the two-layer network architecture extends previous work with similar architectures, adding further support to the distributed nature of working memory representations.

---

## [Referee Report · Reviewer #2 (Public review)]

In this manuscript, Yang et al. present a modeling framework to understand the pattern of response biases and variance observed in delayed-response orientation estimation tasks. They combine a series of modeling approaches to show that coupled sensory-memory networks are in a better position than single-area models to support experimentally observed delay-dependent response bias and variance in cardinal compared to oblique orientations. These errors can emerge from a population-code approach that implements efficient coding and Bayesian inference principles and is coupled to a memory module that introduces random maintenance errors. A biological implementation of such operation is found when coupling two neural network modules, a sensory module with connectivity inhomogeneities that reflect environment priors, and a memory module with strong homogeneous connectivity that sustains continuous ring attractor function. Comparison with single-network solutions that combine both connectivity inhomogeneities and memory attractors shows that two-area models can more easily reproduce the patterns of errors observed experimentally.

Strengths:

The model provides an integration of two modeling approaches to the computational bases of behavioral biases: one based on Bayesian and efficient coding principles, and one based on attractor dynamics. These two perspectives are not usually integrated consistently in existing studies, which this manuscript beautifully achieves. This is a conceptual advancement, especially because it brings together the perceptual and memory components of common laboratory tasks.

The proposed two-area model provides a biologically plausible implementation of efficient coding and Bayesian inference principles, which interact seamlessly with a memory buffer to produce a complex pattern of delay-dependent response errors. No previous model had achieved this.

Weaknesses:

The correspondence between the various computational models is not clearly shown. It is not easy to see clearly this correspondence because network function is illustrated with different representations for different models. In particular, the Bayesian model of Figure 2 is illustrated with population responses for different stimuli and delays, while the attractor models of Figure 3 and 4 are illustrated with neuronal tuning curves but not population activity.

The proposed model has stronger feedback than feedforward connections between the sensory and memory modules (J_f = 0.1 and J_b = 0.25). This is not the common assumption when thinking about hierarchical processing in the brain. The manuscript argues that error patterns remain similar as long as the product of J_f and J_b is constant, so it is unclear why the authors preferred this network example as opposed to one with J_b = 0.1 and J_f = 0.25.

---

## [Referee Report · Reviewer #3 (Public review)]

Summary:

The present study proposes a neural circuit model consisting of coupled sensory and memory networks to explain the circuit mechanism of the cardinal effect in orientation perception which is characterized by the bias towards the oblique orientation and the largest variance at the oblique orientation.

Strengths:

The authors have done numerical simulations and preliminary analysis of the neural circuit model to show the model successfully reproduces the cardinal effect. And the paper is well-written overall. As far as I know, most of the studies on the cardinal effect are at the level of statistical models, and the current study provides one possibility of how neural circuit models reproduce such an effect.

Weaknesses:

There are no major weaknesses and flaws in the present study, although I suggest the author conduct further analysis to deepen our understanding of the circuit mechanism of the cardinal effects.

---

## [Author Response]

The following is the authors’ response to the previous reviews.

**Reviewer #3:**
I appreciate the revisions made by the author which address all of my concerns.Nevertheless, I have some new questions when I read the paper again. These questions are not necessarily criticisms of the paper, which may reflect the gap in my understanding. Meanwhile, it also reflects the writing might be improved further.- Fig. 1:I understand that a critical assumption for generating the required result is that the oblique orientation has lower "energy" than the cardinal orientation (Fig. 1G). Meanwhile, I always have a concept that typically the energy is defined as the negative of log probability. If we take the log probability plotted in Fig. 1A, that will generate an energy landscape that is upside down compared with current Fig. 1G. How should I understand this discrepancy?

As the reviewer pointed out, a higher prior distribution near cardinal orientations causes cardinal attraction in typical Bayesian models, which can correspond to lower energy around these orientations. Additionally, in the context of learning natural statistics, Hebbian plasticity in excitatory connections strengthens recurrent connections and drives attraction toward more prevalent stimuli within neural circuits.

However, as demonstrated by Wei and Stocker (2015), Bayesian inference model can also produce cardinal repulsion when optimizing encoding efficiency. In our network, this efficient encoding is achieved through heterogeneous lateral connections and inhibitory Hebbian plasticity in the sensory module, resulting in lower energy near oblique orientations. Thus, the shape of prior distribution does not have a direct one-to-one correspondence with the bias pattern or the dynamic energy landscape.

- Fig. 3 and its corresponding text.I understand and agree the Fig. 3B&C that neurons near cardinal orientations are shaper and denser. But why the stimulus representation around cardinal orientations are sparser compared with the oblique orientation? Isn't more neurons around cardinal orientation implying a less sparser representation?

Indeed, with sharper tuning curves, having more neurons can result in a sparser representation. Consider an extreme case where each orientation, discretized by 1°, is represented by only one active neuron with a tuning width of 1°. While this would require more neurons to represent overall stimuli compared to cases with wider tuning curves, each stimulus would be represented by fewer neurons, aligning with the traditional concept of sparse coding.

However, in Fig. 3 and corresponding text, we did not measure the sparseness of active neurons for each orientation. Instead, we used the term ‘sparser representation’ to describe the increased distance between representations of different stimuli near the cardinal orientations. Although this increased distance can be consistent with the traditional concept of sparse coding, to avoid any confusion, we have revised the term ‘sparser representation’ to ‘more dispersed representation’ in the 3rd paragraph in pg. 5 and the 3rd paragraph in pg. 6.